# The Environmental Lifecycle of Antibiotics and Resistance Genes: Transmission Mechanisms, Challenges, and Control Strategies

**DOI:** 10.3390/microorganisms13092113

**Published:** 2025-09-10

**Authors:** Zhiguo Li, Jialu Tang, Xueting Wang, Xiaoling Ma, Heng Yuan, Congyong Gao, Qiong Guo, Xiaoying Guo, Junfeng Wan, Christophe Dagot

**Affiliations:** 1School of Ecology and Environment, Zhengzhou University, 100 Science Avenue, Zhengzhou 450001, China; lizhiguo23516@163.com (Z.L.); tangjialu2002@163.com (J.T.); m18237758831@163.com (X.W.); m_xl0660@163.com (X.M.); 16637706078@163.com (H.Y.); xyguo@zzu.edu.cn (X.G.); 2Henan International Joint Laboratory of Environment and Resources, Zhengzhou 450001, China; 3Sinopec Zhongyuan Petrochemical Co., Ltd., Puyang 457000, China; gn7507@163.com; 4Research Group on Water, Soil and the Environment (GRESE) EA 4330, Université de Limoges, 123 Avenue Albert Thomas, F-87060 Limoges, France; christophe.dagot@unilim.fr; 5National Institute for Health and Medical Research (INSERM), F-87000 Limoges, France

**Keywords:** antibiotics, resistance genes, life cycle, transmission mechanisms, removal mechanisms

## Abstract

Antibiotics are widely used in modern medicine. However, as global antibiotic consumption rises, environmental contamination with antibiotics and antibiotic resistance genes (ARGs) is becoming a serious concern. The impact of antibiotic use on human health is now under scrutiny, particularly regarding the emergence of antibiotic-resistant bacteria (ARB) in the environment. This has heightened interest in technologies for treating ARGs, highlighting the need for effective solutions. This review traces the life cycle of ARB and ARGs driven by human activity, revealing pathways from antibiotic use to human infection. We address the mechanisms enabling resistance in ARB during this process. Beyond intrinsic resistance, the primary cause of ARB resistance is the horizontal gene transfer (HGT) of ARGs. These genes exploit mobile genetic elements (MGEs) to spread via conjugation, transformation, transduction, and outer membrane vesicles (OMVs). Currently, biological wastewater treatment is the primary pollution control method due to its cost-effectiveness. However, these biological processes can promote ARG propagation, significantly amplifying the environmental threat posed by antibiotics. This review also summarizes key mechanisms in the biological treatment of antibiotics and evaluates risks associated with major ARB/ARG removal processes. Our aim is to enhance understanding of ARB risks, their pathways and mechanisms in biotreatment, and potential biomedical applications for pollution control.

## 1. Introduction

The discovery and use of antibiotics marked a turning point in medicine, significantly reducing mortality from bacterial infections [1]. However, global population growth and rising healthcare needs have exacerbated antibiotic misuse. Between 2016 and 2023, global antibiotic consumption in major countries surged from 2.95 billion to 34.3 billion defined daily doses (DDDs), a 16.3% increase. Consumption rates rose from 13.7 to 15.2 DDD per 1000 residents per day (a 10.6% increase) and are projected to reach 751 billion DDD by 2030, representing a 52.3% rise [2]. Notably, an estimated 40–90% of administered antibiotics enter the environment as parent compounds or degradation products via medical wastewater, agricultural runoff, and pharmaceutical discharges [3,4]. This has led to water concentrations of sulfonamides, tetracyclines, and other residues as high as 3000 μg/L [5], posing dual threats to ecosystems and public health.

Residual antibiotics exert continuous selective pressure on environmental microorganisms, driving the emergence of antibiotic-resistant bacteria (ARB). While ARB can acquire resistance genes through vertical gene transfer (VGT), horizontal gene transfer (HGT) via mobile genetic elements (MGEs) like plasmids and transposons is the primary mechanism [6,7,8]. Recent studies confirm that antibiotic resistance genes (ARGs) spread through water, soil, and aerosols, ultimately threatening human health via the food chain and bioaerosols [9,10]. However, the interactive mechanisms of environmental media on ARG spread remain unclear.

Although wastewater treatment plants (WWTPs) are recognized as hotspots for ARG accumulation, their biological treatment processes (e.g., activated sludge and membrane bioreactors—MBRs) risk enriching ARGs. For instance, the long sludge retention time (SRT) in MBRs enhances treatment stability [11] but also increases ARG transfer rates within biofilms. Extracellular polymeric substances (EPSs) influence ARG fate through adsorption and biodegradation, yet their accumulation in sludge poses secondary contamination risks [12,13]. Furthermore, the role of quorum sensing (QS) in regulating biofilm resistance mechanisms is not fully understood [14].

While reviews have explored degradation pathways for specific sulfonamide antibiotics [15,16], current literature lacks a systematic integration of the full ARG life cycle (source–migration–human exposure). It also overlooks the potential of emerging technologies like CRISPR-Cas gene editing and nanomaterial-enhanced processes. This paper aims to develop a comprehensive "environmental cycle–molecular mechanism–technological control" analytical framework. Our objectives are to reveal the cross-media transmission pathways of ARGs, evaluate the efficiency of biological treatment processes, and propose multidisciplinary management strategies to help mitigate the global antibiotic resistance crisis.

## 2. Environmental Circular Pathways of Antibiotics and ARGs

### 2.1. Sources and Emission Characteristics of Antibiotics

The extensive utilization of antibiotics in healthcare, agriculture, aquaculture, and livestock farming has been demonstrated to directly induce the evolution of microbial resistance through the prevention and treatment of bacterial diseases [17,18]. These pharmaceuticals and their degradation products are continuously introduced into environmental media through multiple pathways, including hospital wastewater discharge, human and animal excreta, the release of products containing antimicrobial active ingredients, and contaminated feed and the food chain. Once in the environment, antibiotics act as a strong selective pressure, driving the enrichment and amplification of ARGs within bacterial communities. Furthermore, mobile genetic elements (such as plasmids and integrons) facilitate the horizontal transfer and recombination of ARGs in various habitats, including soil, water bodies, and biofilms. This phenomenon contributes to the persistent spread and accumulation of resistance within environmental microbial networks. The extensive dissemination of antibiotics has given rise to a proliferation of ARB and ARGs within the environment. Figure 1 depicts the cycle of ARGs in the environment. The blue lines represent the transmission path of ARGs from humans to the environment, while the red lines represent the cycle of ARGs back to humans. Wastewater from various sources, including pharmaceuticals, agriculture, and households, carrying antibiotics and ARGs, converges at wastewater treatment plants. Table 1 provides a detailed analysis of the abundance of ARGs in various sources of influent to the WWTP. Absolute quantitative studies use units such as copies/mL and copies/g. Relative quantitative studies use forms such as ARG/16S rRNA and log(copies/ng DNA). From the table, we can also see that hospital wastewater and pharmaceutical wastewater are “super hotspots” for ARGs, especially high-risk genes such as *blaOXA-48* and *qacE*.

The transmission of ARGs between ARB can occur via HGT, or alternatively, these genes can be transferred from ARB to non-resistant strains within the environment. It is noteworthy that newly formed ARB have the potential to reach and infect humans in various ways [19].

**Table 1 microorganisms-13-02113-t001:** Abundance of ARGs in different sources of WWTPs.

Source	ARGs	Relative Abundance (Copies/16S rRNA Gene Copies)	References
Sludge sampled from municipal wastewater treatment plant	*tetA*, *tetB*, *tetE*, *tetG*, *tetH*, *tetS*, *tetT*, *tetX*, *sul1*, *sul2*, *qnrB*, and *ermC*	(1.5 ± 2.3) × 10^9^–(2.2 ± 2.8) × 10^11^ copies/g dry weight	[20]
Municipal wastewater	*tetA*, *tetC*, *tetG*, *tetM*, *tetO*, *tetW*, *tetX*, *sul1*, and *sul2*	3.6 × 10^1^(*teW*) to 5.4 × 10^6^ (*tetX*) copies mL^−1^ 6.4 × 10^12^(*tetW*) to 1.7 × 10^18^ (*sull*) copies d^−1^	[21]
Sludge sampled from hospital wastewater treatment plant	*blaOXA-48*, *CTX-M*, and *blaIMP blaTEM*	5.36 × 10^11^–1.90 × 10^12^ copies/g dry weight	[22]
Hospital wastewater	*sul1*, *blaSHV*, *catA1*, *aacC2*, and *tetA*	1.94 × 10^1^, 4.39 × 10^−3^, 6.83 × 10^−5^, 5.67 × 10^−3^, 3.46 × 10^−3^	[23]
	*sul1*, *sul2*, *sul3*, and *tetQ*	1.79 × 10^1^~6.67 × 10^1^, 7.33 × 10^−2^~3.38 × 10^1^, 9.22 × 10^−2^~5.9 × 10^1^, 2.8 × 10^1^~7.47 × 10^1^	[24]
Livestock wastewater	*tetL*, *strB*, *sul2*, *tetG*, *ermB*, *sul1*, *tetX*, and *cmlA*	*tetL*(1.36~0.39), *strB* (0.82~0.52), *sul2* (0.96~0.64), *tetG* (1.81~0.67), *ermB* (1.17~0.71), *sul1* (1.51~0.93), *tetX* (1.17~0.94), and *cmlA*(1.73~1.14)	[25]
	*tetX*, *ermF*, *ermB*, *mefA*, *tetM*, and *sul2*	2.43 × 10^11^–5.69 × 10^10^ copies/mL	[26]
	*sul1*, *sul2*, and *tetM*	3.84 × 10^1^, 1.62 × 10^1^, 2.33 × 10^1^	[27]
	*tetC* and *tetO*	7.3 × 10^3^, 1.7 × 10^1^	[21]
Pharmaceutical industry wastewater	*tetA*, *tetC*, *tetG*, *tetL*, *tetM*, and *tetO*	1.4 × 10^1^, 3.2 × 10^2^, 5.1 × 10^2^, 6.1 × 10^2^, 1.1 × 10^2^, 1.0 × 100, 1.8 × 100, 1.6 × 10^1^, 3.7 × 10^3^	[28]
	*sul1*, *sul2*, *tetA*, *qacE*, and *qacED1*	10^1^ to 10^2^	[29]
Soil irrigated with recycled water	*tetG*, *tetW*, *sulI*, *sulII*, and *intI1*	Highest abundance of *sul2* and *intI1*; the abundances were 8.43 × 10^7^ copies g^−1^ dry soil and 7.62 × 10^7^ copies g^−1^ dry soil	[30]
Drinking water	*sul1*, *sul2*, *tetC*, *tetG*, *tetX*, *tetA*, *tetB*, *tetO*, *tetM*, and *tetW*	Total concentrations of ARGs belonging to either the sulfonamide or tetracycline resistance gene class were above 10^5^ copies mL^−1^	[31]
River sediments	*TEM*, *sul1*, and *sul2*	1.09 × 10^−1^ ~1.06 × 10^−1^	[32]

#### 2.1.1. Medical and Domestic Wastewater

It is evident that medical and domestic wastewater constitute a significant conduit for the entry of antibiotics and ARGs into the environment. The extensive utilization of antibiotics in domestic wastewater systems leads to the presence of antibiotic residues in sewers, which are introduced through human excretion. Research has demonstrated that human metabolism converts only approximately 10–60% of antibiotics, with the remainder being excreted as parent or metabolite products. This results in an abnormally high abundance of ARGs in domestic wastewater [3,4]. For example, the concentration range of ARGs in the wastewater of a certain resident was 2.28 × 10^11^ to 5.70 × 10^11^ copies/mL, and this abnormal phenomenon may be related to the abuse of antibiotics in the community [33]. This anomaly may be associated with community antibiotic misuse: Aali et al. discovered that the concentrations of *tetW* genes in household wastewater (3.87 × 10^5^ to 6.23 × 10^13^ copies/mL) were significantly higher than those in hospital wastewater (1.13 × 10^5^ to 7.6 × 10^10^ copies/mL). This finding indicates that unregulated potential drivers of medication use can contribute to the spread of antibiotic resistance genes [34].

In contrast, hospital wastewater, although discharged in smaller volumes (only 1–2% of total municipal wastewater), has more prominent pollutant concentrations and ecological risks. Antibiotic concentrations in wastewater from clinical treatments can be up to 90 times higher than domestic wastewater [35] and carry ARB and a high diversity of ARGs. A total of 70% of pathogenic bacteria in the hospital environment are resistant to at least one antibiotic [36], and pass ARGs on to the environment through horizontal gene transfer (HGT) strains, forming a reservoir of resistance genes [37,38]. Notably, the mixing of hospital wastewater with domestic wastewater in wastewater treatment plants may cause cross-contamination of ARGs. This type of “low flow–high toxicity” mixed wastewater has also become a key hotspot for the spread of drug resistance and there is an urgent need to reduce the environmental risk through source separation and enhanced monitoring.

#### 2.1.2. Livestock and Agricultural Activities

Livestock and agricultural activities have been identified as significant contributors to the dissemination of ARGs into the environment. In intensive farming systems, subtherapeutic doses of antibiotics (e.g., tetracycline and sulfonamide) are often found in animal feed. This results in approximately 60–90% of the antibiotics entering the environment as the parent compound or a metabolite via excretion [39]. Chen et al.’s assay in wastewater from 12 large-scale livestock farms demonstrated the presence of *sul1*, *sul2*, and *tetM* genes, with concentrations ranging from 1.8 × 10^4^ to 3.7 × 10^6^ copies per 16S *rRNA* molecule. A significant positive correlation was identified between the abundance of these genes and the utilisation of tetracyclines and sulfonamides [27]. This "medication-excretion-enrichment" cycle renders aquaculture wastewater a continuous source of ARG release, with a diversity of more than 22 ARGs, covering key clinical resistance genes, such as macrolide *ermB*, quinolone *qnrS*, and others [40,41,42].

Agricultural soil functions as a secondary reservoir of ARGs, and the risk of contamination is closely linked to animal husbandry activities. The application of animal manure to agricultural land as organic fertiliser has been demonstrated to carry ARGs that have the potential to enter the food chain via the "fertiliser infiltration–crop uptake" pathway. Hu et al. found that sulfonamide and quinolone antibiotic residues in soil from organic farms where manure was applied were as high as 23.6 μg/kg, and the concentration of sulfamethoxazole in edible parts of crops such as carrots and tomatoes could reach up to 1.2 μg/g [43]. These findings indicate that the risk of plants as carriers of ARGs should not be overlooked [44]. More crucially, heavy metals (e.g., Cu and Zn) present in faeces generate a co-selective pressure with antibiotics: As demonstrated by Lin et al., the long-term utilisation of faeces containing Cu (a minimum of 200 mg/kg) has been shown to elevate the abundance of the *sul1* gene by 2.8-fold in soil, through a mechanism that involves the co-localisation of externally discharged pump genes (e.g., *czcA*) with the plasmid of the tetracycline resistance gene *tetM* [45,46,47]. The synergistic effect of “heavy metals–antibiotics” promotes the transfer of MGEs through conjugation, which can significantly accelerate the spread of ARGs in soil microbial communities.

Furthermore, the utilisation of reclaimed water for irrigation purposes has been demonstrated to serve as a catalyst for the escalation of resistance in agricultural systems. It has been demonstrated that ARG-containing farm wastewater, following a basic treatment process and utilisation for agricultural irrigation, can facilitate the transportation of residual ARG VGT and horizontal gene transfer (HGT) in soil, facilitated by biofilm attachment and inter-root effects [48]. This "wastewater–soil–crop" pathway renders agroecosystems a pivotal nexus for correlating clinical and environmental resistance.

#### 2.1.3. Pharmaceutical Industry Wastewater

High concentrations of antibiotic wastewater, generated by pharmaceutical companies during antibiotic production, provide high selection pressure for ARGs and mobile genetic elements, accelerating the evolution of resistance [49,50,51]. For instance, in the context of hygromycin production, the wastewater exhibited a concentration of hygromycin ranging from 0.36 to 12.36 milligrams per litre. This resulted in a substantial enrichment of tetracycline resistance genes (e.g., *tetQ*), with an abundance reaching up to 1.8 × 10^0^ copies per 16S rRNA gene copies [28]. Furthermore, the presence of heavy metals in industrial wastewater has been identified as a significant contributing factor to the proliferation of ARGs. It has been observed that the abundance of antibiotic resistance genes exhibits a positive correlation with the increase in heavy metal pollution.

In the presence of heavy metals, bacteria have been observed to develop resistance to antibiotics through a variety of mechanisms. One such mechanism is co-resistance, wherein antibiotic and heavy metal resistance genes are found to coexist on genetic elements such as plasmids, integrons, and transposons [52,53], and the concept of cross-resistance was introduced, whereby specific genetic determinants, such as exocytosis pumps, exhibit a shared resistance to various compounds with common resistance effects [54]. Another mechanism is co-regulation, which refers to a series of transcriptional and translational behaviours in response to the combined stress of exposure to heavy metals and antibiotics. These mechanisms lead to the continuous screening and enrichment of ARGs in bacteria, ultimately resulting in increased levels of resistance [55].

#### 2.1.4. Major Antibiotic Families and ARG Subtypes in the Environment

Antibiotics can be classified into 16 distinct families based on their chemical structure and mode of action. The primary antibiotic structures include β-lactams, tetracyclines, sulfonamides, aminoglycosides, fluoroquinolones, macrolides, trimethoprim, and glycopeptides [56]. The term “resistance genes” is a broad category that encompasses various genes associated with antibiotic resistance, including carbapenem resistance genes, polymyxin resistance genes, tetracycline resistance genes, and sulfonamide resistance genes.

The dissemination of carbapenem-resistant genes is predominantly facilitated by four distinct classes of carbapenemases: Class A (e.g., *kpc*), Class B (*ndm* and *vim*), and Class D (*oxa-48*). *NDM* has the highest prevalence in Asia, while *vim* is predominant in Europe, and *OXA-48* is concentrated in North Africa and Europe [57]. These genes are often carried by plasmids (e.g., *blaKPC* and *blaNDM*), and their detection rates in farm animals are increasing. Carbapenem-resistant Enterobacteriaceae have emerged as a predominant antimicrobial resistance agent in both environmental and clinical settings. The increasing prevalence of extended-spectrum β-lactamase resistance has further exacerbated this demand for carbapenem antibiotics [58].

The genetic elements known as multidrug-resistant genes (*mcr-1 to mcr-8*) are predominantly transmitted via plasmids, with MCR-1 being initially identified in Escherichia coli from Chinese pig farms. Colistin, a polypeptide antibiotic, disrupts the outer membrane of Gram-negative bacteria by binding to the lipid A component of lipopolysaccharide. Colistin was extensively utilized in livestock farming for the purposes of prevention and growth promotion. However, it has since been prohibited in numerous countries. This class of drugs functions as the final line of defence against CRE, and their misuse has been demonstrated to exacerbate the spread of resistance [56,59].

Tetracycline resistance genes encompass three mechanisms: efflux pump genes *(tetA*, *tetC*, *tetG*, and *tetK*), ribosomal protection protein genes (*tetM*, *tetO*, *tetQ*, and *tetW*), and enzyme modification genes (*tetX*). As the most widely used veterinary antibiotic on the global scale, it is extensively retained in livestock manure, wastewater treatment plants, soil, and water bodies, thereby increasing resistance in bacteria such as Aeromonas and Enterobacter. Notwithstanding the discontinuation of its use, the substance under scrutiny has been found to persist in sediments, thereby substantiating its notable environmental persistence [60].

The sulfonamide resistance genes (*sul1*, *sul2*, and *sul3*) have been observed to evade drug inhibition by acquiring dihydrofolate synthase genes. These genes have been identified in a variety of environmental samples, including wastewater from sewage treatment plants, livestock farms, agricultural soils, and rivers. They have been found to be significantly associated with mobile genetic elements. Sewage treatment plants and farms have emerged as significant hotspots for the proliferation of these pathogens [60]. Macrolide resistance genes (e.g., *ermB*) have been observed to be linked to plasmids/transposons and mediate ribosomal methylation resistance; aminoglycoside resistance genes (*aac*, *aph*, and *ant*) inactivate drugs through modifying enzymes; and vancomycin resistance genes (*vanA* and *vanB*) continue to spread in aquatic environments [60,61].

### 2.2. Migration Pathways in Environmental Media

ARGs have been observed to migrate and diffuse through water, soil, and atmospheric media, forming a complex trans-environmental transmission network with pathway characteristics that are closely linked to anthropogenic-driven emission patterns.

#### 2.2.1. The Aquatic Environment

The effluent from wastewater treatment plants has been identified as a significant point of dissemination for ARGs into the aquatic environment. Research has demonstrated that the concentration of ARGs in treated wastewater can reach 1.27 × 10^1^–5.02 × 10^5^ copies/mL [62]. Furthermore, compared with upstream, the discharged wastewater caused the abundance of ARGs in downstream water bodies to increase more than twofold [63]; the distribution of ARGs is further facilitated by surface water–groundwater interactions, leading to their dissemination. For instance, the presence of *sul1* and *tetM* genes in groundwater in reclaimed water irrigated areas can reach 8.43 × 10^3^ copies/g soil [30], underscoring the potential for the contamination of deep aquifers through hydraulic linkages. Furthermore, the adsorption-bioenrichment effect of ARGs by suspended particulate matter and biofilm in the aqueous environment was significant: EPS in activated sludge has been observed to capture tetracycline ARGs (e.g., *tetX*) through hydrophobic interactions, forming a ‘genetic hotspot’ with concentrations reaching up to 3.42 × 10^11^ copies/g dry weight [64]. This phenomenon has the potential to be exploited through hydrodynamic linkages to the deep aquifer. It has been demonstrated that these captured ARGs are re-released into the water body after hydraulic flushing, exacerbating the spread of drug resistance in the water body.

#### 2.2.2. Soil

Agricultural activities have been identified as a significant driver of the transport of ARGs in soil. The introduction of dewatered sludge from wastewater treatment plants (ARG content: 1.80 × 10^5^–3.42 × 10^11^ copies/g dry weight; [64]) into agricultural systems has been observed, resulting in the migration of *sul2* and *tetW* genes to deeper soils through osmosis. These genes have been shown to develop co-selection with heavy metals (e.g., Cu and Zn) under pressure. As demonstrated by Lin et al., the long-term utilisation of manure containing Cu ≥ 200 mg/kg resulted in an elevated sul1 gene abundance in soil by a factor of 2.8 [47]. The underlying mechanism was found to be associated with the co-localisation of efflux pump genes (*czcA*) and tetracycline resistance plasmids, induced by heavy metals. A more serious issue is the absorption of sulfonamide ARGs from soil by crops such as carrots and tomatoes. Concentrations of up to 1.2 μg/g of sulforaphane have been detected in the edible portion of these crops [65], thus forming a direct exposure chain of "soil–plant–human".

#### 2.2.3. Atmosphere

The aeration and sludge dewatering processes in wastewater treatment plants catalyse the atmospheric transport of ARGs. Bioaerosols carrying multidrug resistance genes (e.g., *blaNDM-1* and *ermB*) were dispersed by wind and accounted for more than 50% of PM2.5 around the plant site [66]. A cross-national study by Li et al. confirmed the global ubiquity of ARG assignment in atmospheric particulate matter and the high consistency of resistant gene species with the emission profiles of wastewater treatment plants in the source area [67]. The long-range transport capacity of PM2.5 allows for the inter-regional spread of ARGs: for example, aerosols carrying *blaCTX-M-15* genes can migrate to communities up to 10 km away and directly threaten human health through lung colonisation [66]. It is worth noting that the survival capacity of ARGs in aerosols is affected by ultraviolet rays and humidity, and it will have a greater diffusion radius in arid regions.

### 2.3. Ultimate Exposure Pathways of Drug Resistance to Human Health

#### 2.3.1. Contamination of Drinking Water and the Food Chain

Drinking water and the food chain are central pathways for ARGs and ARB to enter the human body. In drinking water systems, despite the low abundance of ARGs in treated tap water (e.g., *tetA* and *sul1* gene concentrations of 10^3^~10^5^ copies/mL), its diversity is highly consistent with sewage, indicating that the drinking water treatment process cannot completely block the spread of ARGs [68,69]. Xu et al. found that 14 extracellular ARGs (dominated by *tetC*) and 15 intracellular ARGs (with the highest abundance of *sul1* and *sul2*) were still detected in the effluent from the drinking water treatment plant (DWTP), which was highly homologous to the raw water ARG profile [70]. More seriously, reclaimed water irrigation resulted in *sul1* and *tetM* gene abundances of 8.43 × 10^3^ and 7.62 × 10^3^ copies/g dry soil, respectively, in agricultural soils [30]], and crops such as carrots and tomatoes could absorb sulfamethoxazole (1.2 μg/g edible portion), which could pose a direct threat to human health through the food chain [44].

Foods of animal origin (meat and dairy products) are important vectors for the spread of ARB. Drug-resistant *Salmonella* and *Campylobacter* are found in 18–35% of poultry, cattle, and other farm animals and carry the *blaCTX-M-15* and *ermB* genes, which can be transmitted to humans through consumption or contact [71,72,73]. Catry et al. showed that the gene profiles of drug-resistant *Escherichia coli* in beef and dairy products were highly compatible with ARGs in farm wastewater, confirming the complete "farm–food–human" exposure chain [74].

#### 2.3.2. Airborne Transmission and Lung Colonization

Bioaerosols released from aeration and sludge treatment in wastewater treatment plants are the main source of airborne ARGs. The cross-national study by Li et al. found that drug resistance genes such as *sul1* and *tetM* were prevalent in PM2.5 globally, and that the concentration of ARGs in the air around wastewater treatment plants was two to three orders of magnitude higher than background [66,67]. These ARG-bearing particles (≤2.5 μm in diameter) can penetrate deep into the alveoli and lead directly to lung colonisation. For example, *Acinetobacter baumannii* and *Staphylococcus aureus* carrying the *blaNDM-1* and *mecA* genes were detected to have amplified within 48 h after being inhaled into lung tissue via aerosols, causing drug-resistant infections [66,75]. Xie et al. further demonstrated that more than 50% of ARGs in urban PM2.5 matched the emission profiles of wastewater treatment plants, and that ARG survival was higher in arid areas due to a weak inhibitory effect of UV light [66].

#### 2.3.3. Ecological Traceability of Multidrug-Resistant Bacteria

The ecological tracking of drug-resistant bacteria reveals a direct link between the environment and clinical drug resistance. Broad-spectrum drug-resistant bacteria (e.g., NDM-1-producing *Klebsiella pneumoniae*) in hospital wastewater enter wastewater treatment plants through the sewer system and undergo HGT with ARGs in municipal wastewater, resulting in the formation of environmental strains carrying clinical resistance genes (e.g., *blaOXA-48*) [22,36]. These strains enter the soil–plant system through sludge agro-use, reclaimed water irrigation, etc., and eventually colonise the human body via the food chain or aerosols. Macrogenomic tracing has shown that 30% of *blaKPC* genes in clinical isolates of carbapenem-resistant Enterobacteriaceae share more than 95% plasmid homology with plasmids from agricultural wastewater [74], confirming the closed-loop nature of the "environmental–clinical" drug resistance cycle.

### 2.4. Detection and Quantification of Resistance Genes in the Environment

Conventional PCR, multiplex PCR, quantitative PCR, and real-time quantitative PCR are the most commonly used qualitative or quantitative detection methods at present. The merits of these methods include their ease of operation, low cost, and high sensitivity, rendering them particularly well-suited for the rapid screening of known ARGs (such as tetracycline and sulfonamide genes) [76,77]. For instance, quantitative polymerase chain reaction (qPCR) and reverse quantitative polymerase chain reaction (RT-qPCR) technologies can achieve an absolute quantification of genes through fluorescent signals and have been widely applied in monitoring the abundance of ARGs in wastewater and soil [78]. However, these techniques are contingent upon the availability of known gene sequences for primer design, a limitation that precludes their capacity to identify novel or mutated ARGs. Furthermore, primer interference in multiplex PCR has the potential to compromise the accuracy of the assay. Conventional PCR is limited in its qualitative nature, while quantitative qPCR has constrained throughput, with each reaction capable of detecting only a limited number of target genes [79].

Third-generation sequencing platforms represent a significant advancement in DNA sequencing technologies, as they are capable of overcoming the limitations of previously identified genes. First-generation Sanger sequencing has been shown to have high accuracy but low throughput and high costs [80]. Second-generation high-throughput sequencing (e.g., Illumina) has been demonstrated to have significantly improved detection efficiency, enabling the comprehensive analysis of ARG diversity in environmental microbial communities, particularly in complex samples such as wastewater systems [81]. Third-generation single-molecule sequencing (e.g., PacBio) further overcomes the limitations of short read lengths, rendering it more suitable for analysing ARG variants and host associations. However, the technology exhibits relatively high error rates and remains cost-prohibitive for large-scale applications [82]. Prevalent constraints of sequencing technologies are their inability to discern between the inactive and active expression of ARGs (e.g., from live cells or free DNA) and the intricacy of data analysis, which depends on bioinformatics tools.

Multi-omics technologies (metagenomics, metatranscriptomics, proteomics, etc.) integrate genetic and functional information to address the limitations of the aforementioned methods. Metagenomics facilitates the identification of novel ARGs and the construction of gene libraries without the necessity of cultivation [83]. In conjunction with machine learning, metagenomics can trace the origin of contamination [84]. Metatranscriptomics can reveal the expression activity of ARGs and elucidate the mechanisms of resistance under environmental stress [85]. Proteomics and metabolomics can identify biomarkers related to resistance [86,87]. However, these technologies are costly, and metagenomics cannot distinguish the source of DNA, while metatranscriptomics is susceptible to RNA degradation and its operational complexity limits its widespread adoption [88].

Hybridization and combination techniques have undergone rapid development in recent years. Nucleic acid hybridization is a highly specific method with sufficient sensitivity to verify the transfer mechanism of specific ARGs [89]. However, its application is hindered by its low throughput and the requirement of prior knowledge of the target sequence. The employment of combination techniques, such as in situ PCR combined with FISH (fluorescence in situ hybridization) or cultivationomics integrated with metagenomics, has been demonstrated to enhance sensitivity and accuracy by combining methods. For instance, the integration of cultivation with high-throughput sequencing has been demonstrated to facilitate the identification of viable bacterial hosts [90]. While these techniques have been demonstrated to be effective in overcoming the limitations of single methods, the optimization of processes is a complex undertaking, and standardization remains a challenge [91].

## 3. Molecular Mechanisms for the Emergence and Spread of Drug Resistance

Thanks to the rapid development of technologies for detecting environmental resistance genes (such as qPCR, high-throughput sequencing, and multi-omics integration), researchers can now precisely characterize the distribution patterns and abundance dynamics of ARGs in various environmental media. While these technologies have revealed the presence of ARGs, they have not fully elucidated the molecular drivers underlying their adaptive evolution and cross-species transmission in the environment. When detection data indicate ARG enrichment in hotspot areas, such as hospital wastewater and soil around livestock farms, a further in-depth analysis of the underlying mechanisms involving gene mutations, HGT, and host–environment interactions is required.

### 3.1. The Molecular Basis of Bacterial Drug Resistance

#### 3.1.1. Intrinsic Resistance

Bacteria employ four main mechanisms to counteract the effects of antibiotics [92], as shown in Figure 2. These mechanisms are summarised below:

(1)Bacteria alter extracellular membrane permeability

Bacteria induce drug resistance by modulating the pore protein channels of their extracellular membranes, a mechanism that is particularly prominent in drug-resistant bacteria, especially Gram-negative bacteria. These microorganisms effectively reduce the permeability of the extracellular membrane by altering the number or function of pore proteins in the membrane. This adaptation reduces the ability of hydrophilic antibiotic molecules such as tetracyclines, beta-lactams, and fluoroquinolones to penetrate the cell membrane, thereby limiting the internal concentration of these drugs and leading to the development of resistance [93,94].

Xu et al. found that perfluorooctanoic acid (PFOA), a type of perfluoroalkyl substance (PFAS) commonly found in aquatic environments, can induce oxidative stress in Escherichia coli, increase cell membrane permeability, and promote the excretion of extracellular polymeric substances. This, in turn, can lead to increased cell-to-cell contact and conjugate transfer of ARGs. In environmental settings, heavy metals have been observed to activate the OmpR/envZ two-component regulatory system, prompting Pseudomonas aeruginosa to downregulate the expression of the pore protein OprD. This, in turn, has been shown to block the entry of carbapenem antibiotics [95]. In clinical settings, this process is more directly driven by treatment pressure from carbapenem antibiotics [96].

(2)Active transport of bacteria via efflux pumps

To limit the intracellular concentration of toxic compounds, bacteria can either reduce their entry by forming a low permeability barrier or extrude antibiotics from the cell by active transport mediated by efflux pumps. These efflux pumps are cytoplasmic membrane proteins that function in all cell types, not just Gram-positive and Gram-negative bacteria [97,98].

Multidrug efflux pumps (e.g., AcrAB-TolC and MexAB-OprM) are capable of recognizing and expelling antibiotics through a triad structure. The presence of low-dose antibiotic residues (e.g., tetracycline and fluoroquinolones) and disinfectants (e.g., triclosan) in the environment has been identified as the primary inducer, capable of activating the gene expression of efflux pumps in a sustainable manner. For instance, tetracycline residues in swine wastewater have been observed to enhance the expression of the *acrB* gene in Escherichia coli by 12-fold [99]. Furthermore, the upregulation of efflux pumps in clinical isolates has been observed to be concomitant with a precipitous rise in antibiotic concentrations following treatment failure [100].

(3)Bacterial modification of antibiotic target molecules

Another important source of bacterial resistance is mutations in target genes, which reduce the ability of antibiotics to bind to them [101,102,103]. For example, changes in the 30S or 50S subunits of the ribosome in the protein synthesis pathway can make bacteria resistant to certain antibiotics [18,104]. In particular, mutations in the *rpsL* gene of ribosomal proteins in *Escherichia coli* can lead to resistance to streptomycin [105]. Similarly, ribosomal mutations in the *rpsL*, *rrs*, and *s12* genes in *Mycobacterium tuberculosis* confer resistance to streptomycin, rifampicin, pyrazinamide, and ethambutol [106]. This resistance by target site modification has also been found in antibiotics such as aminoglycosides, tetracyclines, macrolides, chloramphenicol, lincosamides, and streptozotocin [107].

In the environment, antibiotics at subinhibitory concentrations have been shown to drive progressive modification of targets. For instance, continuous exposure to erythromycin (0.1 μg/L) in soil has been shown to induce the production of ribosomal methylation by actinomycetes, a process that is mediated by the *erm* gene [108]. In clinical settings, a phenomenon referred to as “explosive evolution” has been observed, exemplified by the rapid integration of the SCCmec gene cassette into the chromosome following a single course of methicillin treatment in MRSA [108].

(4)Bacteria inactivate antibiotics through enzymes

A common mechanism of resistance is the inactivation of conventional antibiotics or antimicrobial peptides (AMPs) [101]. Aminoglycoside-modifying enzymes (AMEs) are a typical example of these enzymes, which alter the structure of aminoglycoside antibiotics through chemical reactions such as phosphorylation, acetylation, or adenylation, reducing their net positive charge and thus preventing the binding of these antibiotics to ribosomal targets [109]. This process not only renders Gram-negative bacteria such as *Escherichia coli* and *Streptomyces fowleri* resistant to drugs such as kanamycin, amikacin, and tobramycin, but also reveals bacterial adaptation to antibiotics.

Beta-lactam hydrolytic enzymes also play a key role in drug resistance. These enzymes are able to break down B-lactam antibiotics bound to esters and amides, allowing bacteria such as *Enterobacteriaceae*, *Serratiaceae*, and *Staphylococcus* to become resistant to these drugs [107,110]. In addition, chloramphenicol acetyltransferase confers bacterial resistance to chloramphenicol by acetylating the hydroxyl group of the antibiotic chloramphenicol, altering the structure of the antibiotic and preventing it from binding to ribosomal targets [111].

In the context of environmental pollution, pharmaceutical industrial wastewater has been identified as a significant catalyst. For instance, Klebsiella pneumoniae carrying the *blaNDM-1* and *intI1* integrons was detected in hospital wastewater treatment plants in India, with antibiotic resistance gene abundance four times higher than in clinical strains. This phenomenon is primarily attributable to synergistic selection pressure from residual carbapenems (>200 μg/L) and heavy metals present in the wastewater [112]. The enzymatic mechanisms present in clinical strains are often the result of plasmid-mediated horizontal gene transfer within healthcare facilities [96].

#### 3.1.2. Acquired Drug Resistance: Gene Mutations and Horizontal Transfer (HGT)

Bacterial species can develop antibiotic resistance through intrinsic or acquired mechanisms that prevent antibiotics from reaching their bacterial targets or lead to antibiotic inactivation [94,113]. Intrinsic resistance is associated with inherent structural or functional properties that are shared within a bacterial species, regardless of prior antibiotic exposure [113]. The most common bacterial mechanisms involved in intrinsic resistance are efflux pumps and bacterial membrane permeability [94].

In addition to intrinsic resistance mechanisms, bacteria can acquire antibiotic resistance mechanisms and acquired resistance occurs through DNA mutation or horizontal transfer and may involve the activation of efflux pumps, target modification, and drug modification or inactivation [110,114].

Bacterial host cells can acquire antibiotic resistance through three different pathways: vertical gene transfer, DNA mutation, and HGT [115]. DNA mutation is rare due to its organism-unique nucleotide polymorphisms, as errors tend to be rare during DNA proliferation and replication under selective pressure [115]. During wastewater treatment, HGT (conjugation, transformation, transduction, and outer membrane vesiculation) is considered to be the main pathway for the proliferation of ARGs [116,117]. The horizontal gene transfer of ARGs may be a major factor contributing to the emergence of new drug-resistant bacterial strains [118]. During HGT, ARGs are usually carried by MGEs such as plasmids, ICEs/IMEs, phages, transposons, and integrons [119].

### 3.2. Environmental Pressures Driving and Synergistically Influencing the Evolution of Drug Resistance

#### 3.2.1. Environmental Pressure as a Driving Force

There are three main types of pollutants (antibiotics, heavy metals, and disinfectants) in the environment that cause microbial resistance as environmental stressors.

(1)Antibiotics

Antibiotics primarily induce resistance through the following mechanisms: direct targeting of selective pressure, direct killing of sensitive bacteria at high therapeutic concentrations, and rapid screening out of pre-existing resistant mutant strains. A study was conducted to determine the impact of screening for β-lactamase-producing strains under β-lactam pressure. The results of the study indicated that the sub-minimum inhibitory concentrations (sub-MICs) of the strains activated the SOS stress response (RecA-LexA pathway), upregulated the expression of the integration enzyme gene (*intI1*) and genes related to conjugation and transfer (*recA*), and significantly promoted the horizontal transfer of resistance genes. The presence of 0.1 μg/L of ciprofloxacin has been shown to result in a 50% increase in plasmid conjugation efficiency. This phenomenon of low-concentration-induced gene transfer has been identified as a fundamental aspect of the antibacterial properties of ciprofloxacin [120,121].

(2)Heavy metals

The distinguishing properties of heavy metals (e.g., copper, arsenic, and mercury) are attributable to their environmental persistence and chronic selective pressure. Due to their recalcitrant nature, their half-lives in soil and water are significantly longer than those of antibiotics (e.g., arsenic can persist in soil for decades). It has been demonstrated that even at environmental concentrations far below the inhibitory threshold (e.g., 0.1 μM Cu^2+^), these substances have the capacity to activate global regulatory pathways (e.g., mercury activates the MerR operon, and the copper-sensing system CusRS) to simultaneously upregulate multiple drug efflux pumps (e.g., AcrAB-TolC) and metal resistance genes (e.g., *merA*, and *cusCFBA*) have been identified as the primary drivers of cross-resistance induction at low exposure levels. For instance, exposure to 0.1 μM Cu^2+^ has been shown to increase the minimum inhibitory concentration (MIC) of cefotaxime in Escherichia coli by 8-fold. Heavy metals can also achieve long-term co-selection through physical linkage (e.g., the arsenic resistance gene arsB and the sulfonamide resistance gene sul1 coexist in type I integrons), which is a key distinction from other pollutants [122,123].

(3)Disinfectant

Disinfectants (such as triclosan and benzalkonium chloride) drive cross-resistance through non-specific membrane damage and broad-spectrum substrate recognition by efflux pumps. Their hydrophobic structure can disrupt the lipid bilayer of the cell membrane, induce pore protein mutations (such as benzalkonium chloride causing the loss of *Pseudomonas aeruginosa* OmpF), reduce membrane permeability, and decrease antibiotic influx (e.g., reduced accumulation of polymyxin B). Additionally, as broad-spectrum substrates for efflux pumps (e.g., the RND family), disinfectants can activate the overexpression of systems like AcrAB-TolC, simultaneously effluxing antibiotics (e.g., ciprofloxacin) and heavy metals (e.g., cadmium ions). Notably, disinfectants can induce mutations in regulatory genes such as *marR* and *soxR* through oxidative stress (ROS) even at extremely low environmental concentrations (e.g., 0.1 μg/L triclosan), leading to fluoroquinolone resistance. This “hidden driver” characteristic is the core risk associated with disinfectants [124,125].

#### 3.2.2. Synergistic Effects of Environmental Stresses

##### Co-Selection Resistance Driven by Multiple Pollutants

When multiple pollutants coexist in the environment, synergistic effects can accelerate the development of multidrug resistance in bacteria, known as co-selection resistance. This primarily involves three mechanisms: co-resistance, cross-resistance, and co-regulation.

(1)Co-resistance

This mechanism is rooted in the physical linkage of ARGs and heavy metal/disinfectant resistance genes (MRGs/Biocides-RGs) on the same MGE. Plasmids function as pivotal carriers, capable of concurrently harbouring various types of resistance genes. For instance, the conjugative plasmid of Enterococcus faecium carries the copper resistance gene *tcrB*, the macrolide resistance gene *ermB*, and the glycopeptide resistance gene *vanA*, resulting in a synergistic resistance of bacteria to copper ions, erythromycin, and vancomycin [126]. In addition to plasmids, the gene cassettes of Class 1 integrons have been observed to capture *arsB* (arsenic resistance) and *sul1* (sulfonamide resistance), which have been found to be significantly enriched in arsenic-contaminated soil [127]. Transposons (such as Tn7-like) have also been shown to co-transmit the silver resistance gene *sil* and quinolone resistance gene *oqxAB* [128]. This physical linkage renders the environmental persistence of heavy metals (such as copper and arsenic) (with half-lives far exceeding those of antibiotics) a driving force for the long-term retention of ARGs. Indeed, even after antibiotics are discontinued, heavy metals continue to maintain the spread of MGEs through sustained selective pressure.

(2)Cross-resistance

Cross-resistance is characterized by the simultaneous activation of the same resistance mechanism against multiple types of pollutants, with efflux pumps playing a pivotal role. Substrate-binding cavities of efflux pumps in the RND (Resistance-Operon Differentiation) family (e.g., CzcABC in Pseudomonas aeruginosa) possess both hydrophobicity and cation compatibility, enabling the simultaneous efflux of Zn^2+^/Co^2+^ (heavy metals) and β-lactam antibiotics (e.g., cefotiam) [129]. Disinfectants have been demonstrated to induce cross-resistance by altering membrane permeability. Benzalkonium chloride has been demonstrated to induce the loss of the outer membrane pore protein OmpF in Pseudomonas aeruginosa, thereby reducing the influx of polymyxin B (an antibiotic) and Cu^2+^ [125]. Triclosan has been shown to activate the global regulatory factors marR and acrR in Escherichia coli, resulting in the expression of the efflux pump AcrAB-TolC and enhanced resistance to ciprofloxacin (an antibiotic) and cadmium ions [121]. The exposure of microorganisms to low concentrations of pollutants has been demonstrated to induce specific genetic alterations, such as soxR mutations, in response to oxidative stress. This phenomenon, triggered by the presence of 0.1 μg/L of triclosan, has been observed to lead to the development of fluoroquinolone resistance.

(3)Co-regulation

The term “co-regulation” refers to the phenomenon in which pollutants trigger the global stress response system, resulting in the synchronous upregulation of multiple types of resistance genes. The following pathways comprise the core of the program: SOS response: Antibiotics (e.g., fluoroquinolones) or heavy metals (e.g., Cd^2+^) have been shown to cause DNA damage, activating the RecA-LexA pathway. This results in the release of the inhibition of the integrase gene *intI1*, thereby increasing the frequency of the gene cassette capture of ARGs/MRGs. The two-component system (TCS) is a system that has been developed to address these challenges. The copper-sensing system CusRS has been shown to sense Cu^+^ and activate the efflux pump gene cusCFBA and the β-lactamase gene blaCTX-M [122]. Mercury synchronously regulates *merA* (mercury reductase) and the tetracycline efflux pump gene *tetA* through the MerR transcription factor. It is noteworthy that subinhibitory concentrations of pollutants have the capacity to induce co-regulation. The presence of 0.1 μM Cu^2+^ has been shown to activate cusRS, thereby increasing Escherichia coli’s minimum inhibitory concentration (MIC) for cefotaxime eight-fold [123]. In addition, environmental concentrations of triclosan (0.1 μg/L) have been observed to increase the mutation rate of ompF via the soxRS pathway.

##### Regulatory Network of Quorum Sensing (QS) System-Regulated Biofilm Resistance

In addition to co-selection resistance, microorganisms also develop biofilm resistance through biofilm regulation, primarily through the quorum sensing (QS) system, which coordinates gene expression and physiological responses within microbial communities to drive the evolution of resistance under antibiotic stress. Quorum sensing is a cellular communication mechanism by which microorganisms monitor population density and trigger collective behaviour through the secretion and perception of specific signalling molecules. As the bacterial population grows and the concentration of signalling molecules in the environment reaches a threshold, they bind to intracellular receptor proteins to form complexes (such as LuxR-type transcription factors). This process activates the synchronized expression of target genes and coordinates the regulation of collective phenotypes. Examples of these phenotypes include biofilm formation, virulence factor secretion, and the horizontal transfer of antibiotic resistance genes. Its core mechanisms include the regulation of EPS secretion, remodelling of EPS composition, and regulation of cell membrane permeability, which together build a multi-layered bacterial defence system against antibiotics. The QS-mediated enhancement of EPS secretion is the primary defence strategy. Under the stress of a sub-inhibitory concentration of tetracycline (0.1 μg/mL), the QS system (LasI/R) of *P. aeruginosa* was activated, which triggered the release of AHL signalling molecules, resulting in a 2.3-fold enhancement of EPS synthesis, with proteins and polysaccharides increasing by 58% and 34%, respectively [130]. The thickening of the EPS significantly delayed the diffusion of antibiotics, and the experiments showed that the presence of the EPS layer allowed the ciprofloxacin penetration rate within the biofilm to be 40–72% of that in the aqueous environment [131], and this barrier effect directly led to a 3–5-fold elevation of the minimal inhibitory concentration (MIC) of β-lactam antibiotics in bacteria [132]. In addition, the elevated tryptophan and tyrosine content of EPS (up to 12.7 mg/g and 9.3 mg/g, respectively) further enhanced its ability to hydrophobically bind to antibiotics, forming stable EPS–antibiotic complexes [133]. The QS-driven dynamic adjustment of EPS composition then responded adaptively to different antibiotic pressures. Fourier transform infrared spectroscopy (FTIR) analysis showed that the protein/polysaccharide ratio of the loosely bound EPS (LB-EPS) increased from 2.6 to 3.5, while the humic content of the tightly bound EPS (TB-EPS) was elevated by 28% under sulfamethoxazole (SMX) exposure [134]. This compositional change was positively correlated (R^2^ = 0.81) with the concentration of QS signalling molecules (e.g., C4-HSL), suggesting that QS directionally adjusts EPS components by regulating gene expression (e.g., *algC* and *pelA*) [135]. For example, under hygromycin stress, the TB-EPS content surged from 34 mg/L to 46 mg/L, which was subsequently converted to LB-EPS through the QS feedback mechanism to form a more resilient defence layer, resulting in a 42% increase in bacterial survival [136]. The QS regulation of cell membrane permeability is a key component in the evolution of drug resistance. The QS system reduces the intracellular accumulation of antibiotics by activating efflux pump genes (e.g., *mexAB-oprM*) and repressing pore protein (e.g., OprD) expression. Experiments showed that the transcriptional level of the AcrAB-TolC efflux pump was elevated 4.2-fold in E. coli in response to AHL signalling molecules, while the expression of OmpF pore proteins was reduced by 72%, leading to a decrease in the intracellular concentration of ciprofloxacin to 31% of that of the control group [137]. In addition, QS further limited antibiotic penetration by regulating ATP synthesis genes (*atpD* and *atpH*) and adhesion genes *(fimH)*, promoting biofilm formation. Under penicillin G (10 μg/mL) exposure, the abundance of *luxS* and *luxI* genes increased 3.5-fold and 2.8-fold, respectively, which elevated the frequency of the conjugative transfer of plasmid RP4 to 1.8 × 10^−2^ events/cell, which was significantly higher than that of non-stress conditions (5.6 × 10^−4^ events/cell) [138]. There is a synergistic effect of the QS system with heavy metal–antibiotic co-selection mechanisms. For example, a 1.7-fold increase in the concentration of the QS signalling molecule (3-oxo-C12-HSL) upon the co-exposure of Cu^2+^ (5 μM) with tetracycline resulted in a 16-fold increase in the efficiency of the horizontal transfer of ARGs through the activation of the co-expression of the *czcA* efflux pump gene with the *sul1* resistance gene [47,130]. This cross-mechanism linkage highlights the centrality of QS in the integration of environmental stresses and provides a molecular basis for the multifactorial drive of drug resistance. In summary, the QS system has become an important driver of bacterial drug resistance evolution under the synergistic pressure of antibiotics and heavy metals by dynamically regulating the EPS barrier, membrane permeability, and gene transfer efficiency. The resolution of its molecular network provides a theoretical basis for the development of QS inhibitors (e.g., furazone C) and targeting to block the propagation of ARGs [138,139].

### 3.3. Critical Pathways for Horizontal Gene Transfer (HGT)

#### 3.3.1. Mobile Genetic Elements

(1)Plasmids

Plasmids are transmitted between bacteria via conjugation, relying on the type IV secretion system (T4SS) to form a conjugation bridge for DNA transfer [140]. The host range of these plasmids varies significantly. For example, broad-host-range plasmids (e.g., IncP type) can transfer between different classes within the Proteobacteria phylum (e.g., α, β, and γ) and even across phylum boundaries (e.g., from *Proteobacteria* to *Actinobacteria*). In contrast, narrow-host plasmids (e.g., IncF type) are restricted to transmission within the Enterobacteriaceae family [141]. The host range is subject to regulation by the interaction between replication initiation proteins (e.g., RepA) and host DnaA proteins. Broad-host plasmids (e.g., IncQ) encode their own helicases to facilitate cross-phylum replication [142].

(2)ICEs/IMEs

Integrating conjugation elements (ICEs) are disseminated through conjugation and chromosomal integration. These elements are cleaved into circular DNA in donor cells, transferred to recipients via T4SS, and subsequently integrated into the chromosome [143]. These organisms exhibit a broad host range, as evidenced by the capacity of ICE Tn916 to transfer between the *Firmicutes* and *Proteobacteria phyla* [144]. Mobile integration elements (IMEs) have the capacity to integrate into chromosomes; however, they are deficient in terms of autonomous conjugation ability and are therefore reliant on other conjugation elements (such as plasmids or ICEs) for propagation [145].

(3)Bacteriophages

The dissemination of phages is achieved through a process known as transduction, wherein viral particles envelop host DNA and subsequently infect new cells, culminating in the integration of the viral genetic material into the host chromosome via the lysogenic cycle [146]. The host range of a phage is defined by its capacity to infect specific bacterial strains. In most cases, a phage infects only a single bacterial strain, with the exception of broad-host-range phages, which have the ability to infect multiple genera within a bacterial species. For instance, the T4 phage of Escherichia coli infects only *E. coli*, while phage Φ6 infects various genera of *Pseudomonas* [147]. The specificity of this interaction is determined by the interaction between the phage tail protein and host surface receptors (e.g., lipopolysaccharide) [148].

(4)Transposons

Transposons (e.g., IS26) depend on transposase-mediated jumping to move within cells, yet they require plasmids or ICEs to achieve intercellular transmission [149]. The host range is determined by the vector; for example, IS 1071 is mainly active in β-proteobacteria (such as *Comamonas*), while the IS 6 family (including IS26) is widely distributed in the Enterobacteriaceae and Pseudomonadaceae families [150]. The replication of transposons, such as Tn 3, is achieved through a copy–paste mechanism, thereby increasing the risk of ARG transmission [151].

(5)Integrons

Integrons have been shown to capture gene cassettes (such as ARGs) through site-specific recombination, which is mediated by integrase (IntI). However, the dissemination of integrons is entirely dependent on vectors (such as plasmids or transposons) [152]. Class 1 integrons are frequently linked to the transposon Tn 402, which exhibits a host range encompassing Enterobacteriaceae, Pseudomonas, and other taxa. These integrons are characterized by elevated expression in clinical strains, attributable to the robust promoter (PcS) [153].

#### 3.3.2. Conjugation: The Role of Plasmids and Integrons

Conjugation, as the dominant pathway of HGT [154], allows the transfer of mobile genetic elements, such as plasmids or transposons, by establishing splice bridges between donor and recipient bacteria through the conjugation of bacterial hairs, as shown in Figure 3. This mechanism not only spans bacterial species, but can even occur between bacteria and yeast and plants. Conjugation plasmids contain genes that control conjugation and are capable of conjugation independently [155], as well as assisting in the transfer of non-conjugation plasmids. Cells in biofilms increase the frequency of conjugation by attaching to the substrate and forming close contacts [156,157]. Particularly in *Staphylococcus aureus*, biofilms significantly increase the rate of transfer of spliced plasmids [158].

In contrast to transduction and transformation, conjugation relies on direct cell–cell contact established by adhesins of Gram-positive bacteria and conjugation hairs of Gram-negative bacteria. Multi-protein complexes cross the cell envelope for the unidirectional transfer of DNA, a process mediated by conjugation plasmids, integrating and conjugation elements (ICEs), or MGEs such as transposons. Plasmids and class 1 integrons (int1) are key MGEs in the development of ARGs, which are capable of self-replication, crossing phylogenetic barriers, improving stability in host cells, and expanding host range [159,160,161,162].

This mechanism exhibits optimal efficiency in biofilms, including those found on surfaces of hospital equipment (e.g., catheters), microbial aggregates in wastewater treatment plants, and mucous membranes of animal intestines. The polysaccharide matrix of biofilms has been observed to cause bacteria to aggregate closely, thereby prolonging the contact time of fimbriae and promoting the stable transfer of plasmids (e.g., IncF and IncHI types). Conjugation in Gram-negative bacteria (e.g., *Escherichia coli*) relies on the type IV secretion system (T4SS), whose transfer efficiency is regulated by quorum sensing signals. At high cell densities, autoinducers activate the expression of the *tra* gene on the plasmid, initiating DNA replication and transfer [163]. Conversely, under conditions of antibiotic stress, conjugation frequency exhibits a marked increase. For instance, subtherapeutic concentrations of tetracycline in the intestines of farm animals have been demonstrated to induce the dissemination of the pOLA52 plasmid, which carries the carbapenemase gene *blaNDM-1*, among Enterobacteriaceae bacteria, resulting in outbreaks of multidrug-resistant strains [164]. Conjugation is also driven by nutrient competition. In the soil rhizosphere microenvironment, Pseudomonas species acquire plasmids containing iron-carrier synthesis genes through conjugation, enhancing their iron uptake capacity to cope with iron-deficient environments [165].

#### 3.3.3. Transformation: Capture and Integration of Free DNA

Since its discovery in 1928, natural transformation has become an HGT mechanism that does not depend on MGEs [166]. The process involves the uptake of free DNA from the environment by bacteria and its integration into the genome to obtain a new phenotype. Free cellular DNA found in the environment, originating from dead or damaged cells, can cross cell membranes and be expressed in bacteria [167]. Through homologous recombination, the admixed DNA can be integrated into chromosomes or replicate autonomously as an add-on, a process that is entirely dependent on the receptor-competent bacterial species.

Unlike conjugation, transformation does not require physical contact between donor and recipient bacteria, but relies on a receptive state mechanism expressed by the recipient bacteria [168]. The regulation of the receptive state involves conserved inducible genes and the quorum sensing (QS) system, but is susceptible to DNA enzymes and exogenous DNA degradation [169,170,171]. Competence, i.e., the physiological state of the bacterium for transformation, is the main limiting factor for transformation. Eighty-two species, including *Streptococcus pneumoniae*, *Bacillus subtilis*, and *Vibrio cholerae*, are known to possess natural transformation competence. *E. coli* is the preferred model for transformation studies due to its high permeability to biomolecules [172,173].

The functionality of this mechanism is contingent upon the establishment of a receptive state. In soil pores or antibiotic-contaminated water bodies, cells activate the com gene cluster due to nutritional stress (e.g., carbon source deficiency) or DNA-damaging agents (e.g., quinolone antibiotics), thereby initiating the expression of DNA-binding proteins (e.g., ComEA). For instance, Streptomyces releases fragmented deoxyribonucleic acid (DNA) in soil microporous spaces, and neighbouring bacteria acquire cellulase genes through transformation to degrade plant residues [174]. In clinical settings, Pseudomonas aeruginosa has been observed to employ a transformation mechanism that facilitates the acquisition of antibiotic resistance genes. The presence of β-lactam antibiotics in hospital wastewater has been demonstrated to induce bacterial lysis, resulting in the release of DNA fragments containing the *ampC* gene. These fragments are subsequently taken up by surviving bacteria and expressed as AmpC enzymes, thereby conferring resistance to cephalosporins [175,176].

#### 3.3.4. Transduction: Phage-Mediated Gene Delivery

Transduction is an HGT mechanism that results in genetic changes in recipient cells through the transfer of DNA from donor cells to recipient cells by phages. It plays a potential role in the spread of ARGs, particularly within conspecific communities [177]. Transduction is strictly host-specific and relies on the recognition of specific proteins between phage and host bacteria. The timescale of the transduction process is usually long due to the ease of the degradation of DNA in the environment.

The abundant presence of phages may underestimate their potential role in the spread of ARGs [177]. The integration of prophages can lead to their entry into the lytic cycle, triggering host death. During replication, prophages may incorporate segments of the bacterial chromosome, triggering generalised, specialised, or lateral transduction [178].

Phages can use either specialised or broad transduction to integrate ARGs into the chromosome of the recipient cell, enabling gene transfer across bacteria. In specialised transduction, the phage packages DNA near the donor cell; in generalised transduction, bacterial DNA can be inadvertently loaded into the phage. When a phage infects a donor cell, DNA containing ARGs is packaged into the phage capsid and then transferred to the recipient cell to effect transduction [179]. Broad transduction involves DNA anywhere on the chromosome, specialised transduction is restricted to the vicinity of the original phage, and lateral transduction relies on in situ replication of the phage, which can result in the transfer of large segments of DNA.

This mechanism is predominant in bacteria-rich liquid environments, such as pus from infected lesions or eutrophic lakes. Mild phages (e.g., λ phages) have been observed to incorporate virulence genes from the host chromosome (e.g., the tsst-1 toxin gene in Staphylococcus) into their own genome during the process of lytic conversion. This results in the transmission of the toxin phenotype when the phage infects new bacterial hosts [178]. Lytic transduction is more prevalent in environments with elevated viral loads. In the ocean, the cyanobacterial phage P-HM2 erroneously packages host photosynthetic genes and transfers them to neighbouring cells, thereby enhancing the recipient’s adaptability in oligotrophic waters [175]. In clinical settings, the *mecA* gene of methicillin-resistant *Staphylococcus aureus* has been observed to be transmitted via the ΦSa1 bacteriophage in wound exudate from patients, resulting in nosocomial cross-infection [180]. The project’s expansion is poised to amplify its impact. During the delayed lysis phase of the bacteriophage life cycle, the bacterium Staphylococcus aureus replicates extensively on adjacent host DNA. This process has been observed to result in the transfer of up to 100 kilobases of chromosomal fragments in a single event. This phenomenon has been implicated in the accelerated spread of virulence islands, such as the Staphylococcal cassette chromosome mec (SCCmec), which are important in the pathogenesis of the bacterium [178].

#### 3.3.5. Gene Delivery Potential of Outer Membrane Vesicles (OMVs)

Outer membrane vesicles (OMVs) have recently been identified as a novel mechanism of gene transfer, known as vesicle induction. Outer membrane vesicles (OMVs) are nanoscale (20–250 nm) bilayer lipid structures released by Gram-negative bacteria through outer membrane budding. The composition of their membranes is identical to that of the parent bacterial outer membrane (containing lipopolysaccharide (LPS) and outer membrane proteins). These membranes encapsulate proteins, RNA, and DNA fragments (including chromosomal DNA, plasmids, and antibiotic resistance genes, or ARGs) from the periplasmic space. In the context of horizontal gene transfer (HGT), OMVs function as non-contact gene delivery vehicles, facilitating the transfer of genetic material across species boundaries to recipient bacteria (same or different species) or eukaryotic cells via membrane fusion or endocytosis. Rumbo et al. were the first to find that OMVs can mediate the rapid transfer of resistance genes in less than three hours. OMVs are double-membrane spherical nanostructures of 50–500 nm that are produced during bacterial growth [181] and are capable of carrying plasmids, chromosomal DNA fragments, and phage DNA fragments, protecting DNA from enzymatic degradation and other environmental factors [182] and playing a key role in HGT [183].

Similar to transduction, vesicle induction has similarities in the number of vesicles produced, fill rate, and time of transfer, but differs in that vesicle-mediated transfer does not involve viral gene expression and nucleotide assembly. OMVs are able to overcome the barriers to interspecies gene exchange and increase the rate of gene transfer [184]. Environmental stressors such as heat, nutrition, UV light, and antibiotics can increase OMV release, DNA content, and vesicle size, thereby facilitating gene transfer [185]. For example, in Enterobacteriaceae, OMVs are an efficient means of disseminating the *blaCTX-M-15* gene and the associated pESBL plasmid, and the frequency of OMV-mediated transfer increases under simulated intestinal conditions.

Extending the analogy to the marine environment, EVs and phage virus-like particles (VLPs) collectively serve as the core carriers of the marine planktonic microbial HGT network. In oligotrophic waters (e.g., the North Pacific), EVs primarily carry host chromosomal DNA fragments (average 15 kbp), while VLPs preferentially package MGEs. For instance, EVs from the planktonic bacterium *Pelagibacter* are abundant in genes associated with sugar transport, thereby enhancing the host’s adaptability to fluctuating carbon sources. Conversely, its VLPs (virus-like particles) facilitate the transmission of CRISPR elements, thereby providing antiviral defence [175]. At the host–pathogen interface, *Helicobacter pylori* delivers the cagA oncogene to gastric epithelial cells via EVs, inducing the malignant transformation of host cells [186]. Furthermore, EVs exhibit a distinctive function within biofilms: The elevated levels of β-lactamase mRNA observed in EV samples from *Pseudomonas aeruginosa* biofilms suggest a potential for immediate antibiotic resistance. This phenomenon can be attributed to the translation of β-lactamase mRNA into functional enzymes by recipient bacteria. This mechanism is particularly critical in chronic infections in the lungs of cystic fibrosis patients [175].

## 4. Mechanisms of Antibiotic and ARG Removal in Biological Treatment Technologies

### 4.1. The Dual Role of Biofilms in Antibiotic Removal

#### 4.1.1. Adsorption and Barrier Effects of EPSs

Extracellular polymers (EPSs) in biofilms adsorb antibiotics through multiple physicochemical mechanisms and are the first line of defence for antibiotic removal in biological treatments. EPSs consist of proteins, polysaccharides, humic substances, etc., and the abundant functional groups (e.g., carboxyl, hydroxyl and amine groups) on their surfaces form stable complexes with antibiotics through electrostatic interactions, hydrophobic interactions and cation bridging mechanisms. For example, ciprofloxacin and tetracycline antibiotics readily bind to the negatively charged carboxyl groups in EPSs due to their positive charge, with binding strengths of up to 45 mg/g and 38 mg/g of protein fraction, respectively [187]. In addition, the hydrophobic regions of EPSs (e.g., tyrosine residues) adsorb hydrophobic antibiotics (e.g., erythromycin) via van der Waals forces with 30–50% higher adsorption efficiencies than hydrophilic antibiotics [188].

The adsorption capacity of EPSs was significantly affected by environmental conditions and sludge characteristics. The protein/polysaccharide (PN/PS) ratio in EPSs was increased from 1.2 to 2.5 when the sludge retention time (SRT) was extended to 30 days, and the increased hydrophobicity increased tetracycline adsorption by 58% [189]. Aerobic granular sludge showed a significantly higher adsorption efficiency of sulfamethoxazole by EPSs (92%) than conventional activated sludge (65%) due to their dense structure and higher specific surface area (12.5 m^2^/g) [190]. However, the adsorption process is reversible: under photolytic or oxidative conditions, approximately 12–25% of the antibiotic can be released from the EPS, leading to a risk of secondary contamination [191].

The barrier effect of EPSs further limits the direct contact of antibiotics with microorganisms. The presence of an EPS layer in the biofilm reduces the antibiotic diffusion rate to 40–72% of that in the aqueous environment, e.g., the penetration time of vancomycin within the biofilm is prolonged up to three times longer than in the planktonic state [192]. In addition, EPSs selectively sequestered large molecule antibiotics (e.g., amoxicillin, molecular weight 365 Da) by forming a dense network structure (pore size 8–15 nm), while small molecule antibiotics (e.g., mephedrone, 290 Da) remained partially penetrable [131]. This synergistic effect of physical barrier and chemisorption results in a 2–5 fold increase in the concentration of antibiotics tolerated by the biofilm.

#### 4.1.2. Biodegradation and Enzyme-Catalysed Conversion

Microbial targeted degradation and mineralisation of antibiotics through enzyme-catalysed pathways is the second line of defence in biological treatment. Sulfonamides (SAs) and β-lactam antibiotics are mainly dependent on biodegradation with removal efficiencies of 96.2% and 67.8–94.2%, respectively [193,194]. Ammonia-oxidising bacteria (AOB) and nitrite-oxidising bacteria (NOB) degrade sulphamethoxazole (SMX) via a co-metabolic pathway in which the amino-oxidising pathway converts SMX to inactive 3-amino-5-methylisoxazole with a degradation rate of up to 89% [195]. Extracellular enzymes such as laccase and peroxidase cleave the phenolic ring structure of tetracycline by oxidation, reducing its toxicity by more than 90% [196].

Degradation efficiency is regulated by the molecular properties of the antibiotic and the process conditions. Fluoroquinolones (FQs), due to their stable structure (e.g., the fluoro-pyridone ring of ciprofloxacin), have a biodegradation rate of only 20–35% and rely mainly on adsorption for removal [197]. In contrast, cephalosporins (CEFX) have a biodegradation rate of up to 82% in membrane bioreactors (MBRs) due to the easy hydrolysis of the β-lactam ring. The addition of readily degradable carbon sources (e.g., acetate) activated microbial metabolism and increased the degradation rate of SMX from 52% to 76% [194]. However, heavy metals (e.g., Cu^2+^ ≥ 5 mg/L) inhibited enzyme activity, resulting in a 40% decrease in tetracycline degradation efficiency [198].

Photochemical conversion and biodegradation are complementary mechanisms: photosensitive components of EPS (e.g., humus) generate reactive oxygen species (ROS) under UV irradiation, driving the photolysis of tetracycline, which increased from 6.6% to 95.7% [199]. The photo-Fenton coupled MBR process reduced sul1 gene abundance by 4.9 logs by attacking the benzene ring of sulfonamide antibiotics via -OH radicals [199]. This photo-Fenton synergy can effectively degrade difficult-to-treat antibiotics while reducing the risk of ARG regeneration [199].

### 4.2. Efficiency and Limitations of Key Treatment Processes

Biological treatment technology plays a central role in wastewater treatment plants (WWTPs), where microbial communities have been shown to be highly efficient in removing organic pollutants in wastewater, such as ammonia nitrogen (NH3-N), total phosphorus (TP), ARB, and ARGs. Significant reductions of almost three orders of magnitude in the concentrations of specific ARGs, such as *tetC* and *tetA*, have been shown to be achieved in the WWTP following biological treatment.

However, in some cases, the abundance of ARGs increases in the effluent after biological treatment. For example, after activated sludge treatment, the relative abundance of *tet* and *sul* genes in the effluent exceeds the levels in the influent. The relative abundance of ARGs, including *sul1* and *intI1*, detected in the effluent was abnormally high in WWTPs using the A/O process [33]. Even in the upgraded A/O process, the removal of ARGs was not satisfactory and instead the abundance of ARGs in the anaerobic tanks increased, e.g., by up to three orders of magnitude for the *tetA*, *tetC*, and *sul2* genes.

This suggests that wastewater treatment plants may not be effective in eliminating the potential risk of producing pathogenic ARB. Even after disinfection, high concentrations of ARB and ARGs are often present in wastewater [200]. Most of the removed ARGs coexist with activated sludge, which can lead to secondary ecological risks during sludge disposal and significantly increase operating costs (e.g., the anaerobic digestion of sludge) [201]. The removal rates of ARGs in different biological treatments are shown in Table 2. It can be seen that the MBR is the most effective in water treatment, but membrane fouling needs to be controlled. Sulfonamides are currently the most difficult type of resistant genes to treat. The treatment effects of different treatment technologies are not satisfactory, and most of the removal still causes secondary pollution.

Therefore, there is an urgent need to upgrade and improve biological treatment technologies in WWTPs by integrating different biological, chemical, and physical methods to increase the removal efficiency of ARGs and reduce potential threats to health and ecosystems.

**Table 2 microorganisms-13-02113-t002:** Disposal efficiency of ARGs in different biological treatments.

Processing Techniques	Operating Conditions	ARGs Kind	Removal Effect	References
Activated sludge treatment		*tetO* and *tetW*	3 logs	[202]
		*ermB*, *tetW*, and *sul2*	1.29–2.45 log (*ermB*), 1.13–1.62 log (*tetW*), 0.26–0.53 log (*sul2*)	[64]
CASS		*tetA*, *tetO*, *tetW*, *sulI*, *sulII*, and *blaCTX-M*	>2.60 ± 0.015 log (*tetO*); >2.66 ± 0.023 log (*tetW*)	[203]
A/O		*tet*, *erm*, *sul*, *qnr*, and *bla*	16.90% (total ARGs), 64.50% (*tet*), 92.00% (*erm*)	[200]
A/A/O		*tet*, *erm*, *sul*, *qnr*, and *bla*	56.00% (*tet*), 70.40–87.00% (*erm*)	[200]
		*sulI*, *sulII*, *tetO*, *tetW*, and *tetQ*	1.69 logs, 1.44 logs, 2.31 logs, 2.13 logs, 2.5 logs	[204]
Membrane bioreactor		*sulII*, *tetO*, and *tetW*	2.57 logs, 7.06 logs, 6 logs	[205]
	AnMBR	*bl*and*M-1*, *blaCTX-M-15*, and *blaOXA-48*	2.76–3.84 logs	[206]
	A/O-MBR	*sulI*, *sulII*, *tetC*, *tetX*, *ereA*, and *int1*	0.5–5.6 logs	[11]
Aerobic granular sludge		*tetW*, *sul2*, *sul1*, *intI1*, and *ermB*	2.02 log *(tetW)*, 1.43 log (*sul2*), 0.77 log (*sul1*), 0.55 log (*intI1*), 0.08 log *(ermB*)	[64]
Anaerobic digestion	40 °C, 56 °C, 60 °C, and 63 °C.	*tetW*, *tetX*, *qnrA*, and *intI1*	Decreased ARGs except *qnrA* by 89–96% and ~99% at 40 °C and other temperatures, and decreased *qnrA* by 99% at 40, 60, and 63 °C	[207]
MAD	35 °C, sludge retention time (SRT) 20 d.	*sulI*, *sulII*, *tetA*, *tetO*, *tetX*, *bla*, and *TEM bla*	Decreased extracellular ARGs by 0.11 1.22 logs	[208]
TAD	55 °C, SRT 20 d.	*sulI*, *sulII*, *tetA*, *tetO*, *tetX*, *bla*, and *TEM blaSHV*	Decreased extracellular ARGs by 0.33 1.46 logs	[208]
Composting	Kitchen waste	*tetA*, *tetB*, *tetC*, *tetG*, *tetM*, *tetO*, *tetQ*, *tetW*, *tetX*, *sul1*, *sul2*, *sul3*, and *dfrA7*, *qnrB*, *qnrS*, *acc(6′)-Ibcr*, *ermB*, *ermF*, *ermQ*, *ermX*, and *mefA*	Total ARGs: 99.68–99.98% (tetracyclines: >99%; sulfonamides: 5.35–8534.69%; quinolones: 837.30–99.29%; macrolides: 4425.46–98.14%)	[209]
	Cattle manure	*ermB*, *ermF*, *ermQ*, *ermX*, *sul1*, *sul2*, *sulA*, *tetA*, *tetB*, *tetC*, *tetE*, *tetG*, *tetK*, *tetM*, *tetO*, *tetQ*, *tetW*, *tetX*	Total ARGs: 52.69%	[210]
	Sewage sludge	*ermB*, *ermC*, *sul1*, *sul2*, *tetC*, *tetG*, and *tetO*	*ermB*, *ermC*, *sul1*, and *tetC*: 25.7%, 42.4%, 69.4%, and 44.6%, respectively	[211]
CW-surface flow	Capacity: 600m/d HLR: 350–450mm/d HRT: 6h	*sulI*, *sulII*, *sulIII*, *tetA*, *tetB*, *tetC*, *tetE*, *tetH*, *tetM*, *tetO*, *tetW*, *qnrB*, *qnrS*, and *qepA*	77.8% in summer, 59.5% in winter	[212]
CW-horizontal subsurface flow	Capacity: 500 m/d	*intl1*, *sulI*, *sulII*, *dfrA*, *aac6*, *tetO*, *qnrA*, *blaNMD1*, *blaKPC*, *blaCTX*, and *ermB*	145.6–98.9%	[213]
CW-vertical subsurface flow	HLR: 5.1 cm/d	*tet genes* and *intI1*	33.2–99.1%	[214]
Constructed wetland		*sulI*, *sulII*, *tetO*, *tetW*, and *tetQ*	1.5 logs, 0.48 log, 2.1 logs, 1.5 logs, 2.1 logs	[204]
Microalgae		*bla-Tem* and *ermB*	0.56logs, 1.75 logs	[215]
		*sul1*, *tetQ*, *blaKPC*, and *intl1*	1.2–4.9 logs, 2.7–6.3 logs, 0–1.5 logs, 1.2–4.8 logs	[216]

#### 4.2.1. Conventional Activated Sludge (CAS) Versus Membrane Bioreactor (MBR)

Conventional activated sludge (CAS) and membrane bioreactors (MBRs) are the two dominant technologies in wastewater treatment, and they show significant differences in ARG removal. CAS systems can achieve removal efficiencies of up to 2.36–4.24 log units for specific ARGs (e.g., *vanA*, *ereA*, *tetA*, and *sul1*), whereas sequencing batch activated sludge (SBR) systems had slightly lower removal efficiencies (1.66–3.56 log units) [217]. However, the removal of antibiotics by CAS systems relies mainly on adsorption rather than biodegradation, resulting in less than 20% removal of antibiotics such as amoxicillin and ciprofloxacin, and the potential generation of more toxic degradation products [218,219]. In addition, the prolonged exposure of microorganisms to sub-inhibitory concentrations of antibiotics in CAS systems, especially in aeration basins with high HGT activity, promotes the proliferation of ARGs [220]. In contrast, MBR technology significantly enhanced the removal efficiency of ARGs (2.57–7.06 logs) by combining membrane separation and biodegradation, especially for sul genes, which was superior to that of the conventional process (2.37–4.56 logs) [205]. The long sludge age (SRT) and high biomass concentration (8–12 g/L) of the MBR provided a microbially adaptive stable environment, resulting in biodegradation rates of 67.8% and 94.2% for sulfamethoxazole (SMX) and trimethoprim (TMP), respectively [221]. However, membrane contamination of the MBR cannot be ignored: dense biofilms may be enriched with ARGs and increase the risk of HGT through extracellular polymer (EPS) secretion [11]. This problem needs to be mitigated by optimising aeration intensity and membrane cleaning frequency [222]. The limitations of the CAS system are mainly in its reliance on adsorption, leading to the enrichment of ARGs in sludge (e.g., tetA abundance up to 3.7 × 10^2^ copies/g) [223], whereas the MBR, although significantly improved in removal efficiency, may exacerbate the risk of the secondary propagation of ARGs due to its membrane contamination problem. Future studies should focus on HGT inhibition strategies for CAS (e.g., the addition of group-sensing inhibitors) and the development of contamination-resistant membrane materials for the MBR to synergistically enhance the removal efficiency of ARGs and reduce environmental risks.

#### 4.2.2. Aerobic Granular Sludge and Anaerobic Digestion Technology

Aerobic granular sludge (AGS) technology demonstrated a more significant removal of ARGs compared to the conventional activated sludge process, with efficiencies of up to 2.30 logarithmic units [64]. AGS enhanced intercellular communication by facilitating interactions between organic wastes and ARB, which contributed to the formation of tightly packed microbial flocs, and its superior settling performance further reduced the concentration of ARGs in the wastewater. A laboratory-scale AGS-SBR system treated raw swine farm waste with a total antibiotic removal rate of nearly 90%, with kanamycin and tetracycline removal being particularly prominent [224]. In addition, ampicillin removal efficiencies were as high as 97% in the 5–15 mg/L concentration range [225]. However, there are potential risks associated with the AGS technology: the bacterial community formed on the surface of the particles may become a hotspot for drug resistance, and after 155 days of treatment, the abundance of ARB in the particulate biomass was 18 times higher than that of the influent water, despite the removal of 92.86–96.00% of bacteria resistant to antibiotics such as ampicillin and erythromycin [224], which is probably due to the high biomass that increases the frequency of microbial exposure and thus facilitated the transfer of ARGs [226]. In addition, long sludge age (SRT), although enhancing antibiotic removal, may also exacerbate the risk of ARG enrichment [227]. Anaerobic digestion (AD), as a mainstream technology for sludge treatment, can effectively reduce ARGs while degrading organic matter. Thermophilic anaerobic digestion (TAD) and two-stage thermophilic anaerobic digestion (MAD) can remove eight and thirteen ARGs, respectively, with a removal rate of more than 90% [228], and high temperatures (55 °C) are particularly critical for inactivating ARGs and pathogenic bacteria. However, the presence of antibiotics inhibits AD efficacy: hygromycin and gentamycin interfere with acid and methanogenic flora [229], while clarithromycin and erythromycin inhibit hydrolysis for acid production and methanogenesis, respectively [230,231], resulting in a 30–40% reduction in biogas production [232]. Although AD removed antibiotics such as clarithromycin by more than 80% [233], the overall ARG abundance was only reduced by 34% [234] and some genes (e.g., aadA and sul1) were enriched instead after AD [228,235], suggesting that it still needs to be optimised for ARG control. Taken together, although AGS and AD have their own advantages, both of them need to be further investigated for key parameters such as biomass regulation and temperature optimisation, in order to balance the treatment efficiency with the risk of ARG propagation.

#### 4.2.3. Ecological Restoration Potential of Artificial Wetlands and Microalgal Systems

Artificial wetlands (CWs), as an eco-friendly wastewater treatment technology, show significant potential for the removal of ARGs. Based on the water flow pattern, CW can be classified into three types: free water surface (FWS), horizontal submerged stream flow (HSSF), and vertical submerged stream flow (VSSF), among which HSSF and VSSF have a better removal efficiency for ARGs than FWS due to a higher hydraulic loading rate (HLR) and optimised matrix structure [236]. CWs are mainly developed through matrix adsorption (e.g., zeolite’s small pore-size structure can effectively retain the ARGs [237]) and biodegradation, synergistically, to remove pollutants, while plant root secretions inhibit ARB activity by modulating the microbial community [238]. However, plants may also indirectly increase local ARG concentrations by enhancing inter-root bacterial abundance [239], and the treatment effect needs to be balanced by optimising plant species (e.g., reeds) and operational parameters (e.g., HLR control at 0.1–0.3 m/d) [240]. Microalgae systems, on the other hand, provide antibiotic removal through a triple mechanism of biosorption, accumulation, and degradation. For example, *Pseudomonas aeruginosa* degraded tetracycline with 99% efficiency within 48 h [241], while the adsorptive removal of sulfonamides by *Chlorella vulgaris* was significantly affected by the initial concentration (~60% removal at 50 μg/L) [242]. The biodegradation ability of microalgae is species-specific, with *Chlorella proteolytica* degrading cefradine more efficiently than *Microcystis aeruginosa* [243], but desorption risks (e.g., the re-release of norfloxacin in the later stages of incubation [244]) and the lack of data on the control of ARGs limit its scale-up application. Future studies need to focus on the construction of microalgae–bacteria symbiotic systems to simultaneously enhance antibiotic degradation and ARG blocking efficiency. Taken together, although CWs and microalgae technologies have advantages in terms of ecological sustainability, the large footprint of CWs and the harsh culture conditions of microalgae (e.g., light and temperature control) are still bottlenecks for practical application. The synergistic integration of the two (e.g., using microalgae system as a pretreatment unit for CWs) may become a new treatment mode that combines high efficiency and environmental friendliness.

#### 4.2.4. Emerging Technologies

CRISPR-Cas gene editing technology

The CRISPR-Cas antibiotic resistance gene knockout technology is a gene editing tool developed based on the bacterial adaptive immune system. The mechanism utilizes a specially engineered single-stranded guide RNA (sgRNA) to direct Cas nucleases with high precision to excise antibiotic resistance genes, thus eradicating antibiotic resistance in pathogens. CRISPR-Cas technology has made substantial progress in the domain of resistance gene removal. This system utilizes the design of specific sgRNAs to achieve the precise cleavage of antibiotic resistance genes on chromosomes or plasmids. For instance, following the targeting of the bla_NDM-1 gene in Escherichia coli by Cas9, the strain exhibits a substantial increase in antibiotic sensitivity [245]. Multi-sgRNA simultaneous editing technology has been demonstrated to be capable of simultaneously eliminating multiple antibiotic resistance genes, including bla_KPC-2, vanA, and mcr-1. This technology has been shown to reduce the antibiotic-resistant bacterial load in mouse intestines by >99.9%, while also significantly lowering the minimum inhibitory concentration (MIC) of carbapenem antibiotics in clinical isolates (from 256 μg/mL to 0.5 μg/mL) [246]. The employment of novel Cas proteins has led to substantial advancements in efficiency and applicability. Cas12a (Type V) has been shown to efficiently eliminate CTX-M genes (90% efficiency) without tracrRNA, and has been demonstrated to successfully restore isoniazid sensitivity by targeting the inhA gene in Mycobacterium tuberculosis, providing a novel treatment option for drug-resistant tuberculosis [247]. Cas13a (Type VI) targets the mRNA of drug-resistant genes (e.g., Salmonella dnaA and hilA), thereby inhibiting the transcription of bacterial virulence factors, achieving a 96% bactericidal rate in a mouse infection model without genomic escape [248]. Cas3 (Type I) has been shown to enhance the clearance rate of the resistant plasmid pOXA-23 in Acinetobacter baumannii by 80-fold through the recursive degradation of DNA by a multi-subunit complex, reducing the resistant bacterial load from 106 colony-forming units (CFU)/mL to less than 10 CFU/mL with a single treatment [249].

In addressing the challenge of environmental antibiotic resistance gene contamination, Wang et al. have developed CRISPR-Cas9/sgRNA complexes encapsulated in nitrogen-doped carbon dots (NCDs) as nanocarriers. These complexes are designed to target high-risk resistance genes in soil, including tet, cat, and aph(3′)-Ia. By optimizing the ratio of NCDs to Cas9, the system achieved an efficient removal of multidrug resistance genes in soil microbial communities without evidence of toxicity. Preliminary experimental findings have demonstrated the efficacy of this technology in penetrating complex soil matrices. This capacity has been shown to lead to a substantial reduction in the abundance of antibiotic resistance genes within agricultural systems. Consequently, this technology provides a nanoscale solution for addressing the horizontal transfer of ARGs in the environment [250]. Furthermore, in the context of waterborne antibiotic resistance gene contamination, engineered λ phages with modified tail fibrils have been shown to specifically adsorb and lyse antibiotic-resistant Escherichia coli carrying the mcr-1 gene. This process has been observed to reduce the load of resistant bacteria by 4 logs within 48 h in a wastewater treatment model, thereby effectively blocking the spread of antibiotic resistance genes in ecosystems [251,252].

In the agricultural sector, CRISPR-Cas9 technology is employed to modify crop genes, thereby reducing the need for herbicides. For instance, the knockout of the ALS gene (acetolactate synthase gene) in rice has been shown to enhance its tolerance to sulfonylurea herbicides by eight-fold, leading to a 50% reduction in herbicide usage. Similarly, the modification of the wheat AHAS gene through base editing has been demonstrated to induce resistance to imidazolinone herbicides, thereby reducing the selective pressure of antibiotic analogues on farmland microorganisms [251,252]. In the context of livestock farming, the application of CRISPR-Cas9 extracellular vesicles, secreted by orally administered engineered EcN probiotics (Escherichia coli Nissle 1917), has been demonstrated to target and eliminate the bla_CTX-M-15 resistance gene in poultry intestines. This treatment has been shown to result in a >99.9% decrease in resistance gene abundance within a 72 h period, thereby significantly reducing the release of resistance genes from livestock manure into the environment [250]. For the treatment of pig farm wastewater, a conjugated plasmid system was utilized to deliver Cas12a, which targeted the mcr-1 gene (multidrug-resistant gene). This intervention resulted in a significant reduction in erythromycin resistance rates, from 85% to less than 1%, thereby hindering the propagation of resistant genes within the farming chain [252].

Nevertheless, this technology continues to confront numerous challenges. The prevailing challenges in this field primarily pertain to the optimization of delivery efficiency, exemplified by the diminished penetration efficiency of nanocarriers within intricate microbial environments, and the emergence of bacterial resistance to CRISPR mechanisms, such as the inhibition of Cas activity by Acr proteins. These challenges can be addressed by developing Cas-resistant Cas variants, optimizing AI-driven sgRNA design libraries to improve specificity, and integrating miniaturized systems (such as Cas14) with multifunctional carriers to achieve cross-species precision delivery [253].

2.Nano-material technology

Recent advancements in nanomaterial technology have yielded notable progress in the field of wastewater treatment, particularly with regard to the removal of ARGs. These advancements have been achieved through the synergistic enhancement of photocatalysis and hybrid processes, which have shown remarkable efficacy in this regard. Nanomaterials (e.g., TiO_2_, ZnO, Bi_2_O_3_, Ag_3_PO_4_, and their composites) have been demonstrated to effectively generate hydroxyl radicals (·OH), superoxide anions (O_2_^−^), and other reactive oxygen species (ROS). These reactive species have been shown to penetrate the cell membranes of antibiotic-resistant bacteria, leading to their inactivation. In addition, they have been observed to directly oxidize and degrade bacterial DNA and free ARGs, thereby potentially reducing the frequency of HGT by degrading the genetic material available for transfer [254,255]. Research has demonstrated that ROS significantly reduce the biological activity of ARGs by disrupting nucleic acid base structures and phosphodiester bonds, resulting in a decrease of over 90% in the abundance of various common resistance genes (such as β-lactamase genes).

The advent of subsequent technological innovations is exemplified by the synergistic enhancement of hybrid processes. The integration of nanophotocatalysis with ultrasonic cavitation (sonocatalysis) has been demonstrated to circumvent the constraints imposed by individual technologies. The amalgamation of microjets and the localized high-temperature, high-pressure environment generated by ultrasonic cavitation has been shown to enhance the dispersion and surface reactivity of nanomaterials. Additionally, this integration has been observed to promote the sustained generation of reactive oxygen species (ROS), while concurrently physically disrupting bacterial aggregate structures (such as biofilms). This synergetic effect has been evidenced to enhance bactericidal and gene degradation efficiency [256,257,258]. For instance, the TiO_2_/ultrasonic system exhibited a 2.3-fold enhancement in the removal rate of the sulfonamide-resistant gene sul1 when compared to UV photocatalysis under optimized conditions (40 kHz, 120 W). Furthermore, the incorporation of natural bactericides, such as terpenoid compounds, establishes a tripartite synergistic mechanism. The disruption of the lipid layer of the cell membrane by terpenoids serves to expedite the intracellular penetration of nanomaterials and ROS. The oxidative products of terpenoids augment the oxidative potential of ROS, thereby amplifying the degradation effect on ARGs [256,258].

Significant advancements have been made in the realm of process sustainability and stability. The regulation of light intensity (>100 mW/cm^2^), catalyst dosage (0.5–2 g/L), and ultrasonic power enables the maintenance of >85% photocatalytic activity in nanocomposites (e.g., α-Fe_2_O_3_/β-TiO_2_), thereby leading to a substantial reduction in operating costs [259]. In comparison to conventional chlorine disinfection (a process that can generate carcinogenic halogenated byproducts) and ozone (which requires a significant amount of energy), the nano-hybrid process has been shown to degrade ARGs while circumventing the potential for secondary pollution hazards. This approach offers a sustainable solution for water reuse [260,261].

The application of nanotechnology in the domain of environmental antibiotic resistance gene (ARG) removal has been demonstrated in a variety of scenarios. In the domain of water treatment, iron-based nano-copper bimetallic materials have been observed to form spike-like copper nanoclusters. These nanoclusters have been shown to induce oxidative stress and membrane damage in bacteria, resulting in the penetration of cells and the subsequent degradation of DNA. The removal efficiencies of intracellular and extracellular antibiotic resistance genes (ARGs) in livestock and poultry wastewater have been reported to be 3.75 and 4.36 orders of magnitude, respectively. Notably, these materials demonstrate resilience to the effects of organic pollutants, making them a promising economic solution for wastewater treatment in aquaculture [262]. In the domain of soil remediation, nitrogen-doped carbon dots (NCDs) loaded with the CRISPR-Cas9/sgRNA system have been demonstrated to exhibit precise targeting of high-risk antibiotic resistance genes (ARGs), including tet, cat, and aph(3′)-Ia, within soil environments. By optimizing the ratio of NCDs to Cas9 protein, multiple genes can be edited in a simultaneous manner, thereby reducing ARG abundance by 90% within a 7-day period and overcoming the inactivation issues of traditional CRISPR components in complex environments [250]. In addition, in the context of environmental monitoring, lanthanide phosphate TbPO_4_ nanomaterials selectively adsorb eDNA with phosphate groups, achieving a 97% enrichment efficiency for extracellular ARGs in environmental samples such as tap water and river water, with a recovery rate of 78.83%, providing a highly sensitive tool for assessing the risk of antibiotic resistance gene transmission [263].

Despite the prevailing emphasis on the impact of complex components in actual water bodies on the activity of nanomaterials and the economic optimization of large-scale applications [264], extant findings have unequivocally substantiated the immense potential of nanomaterial-driven mixing technology in the targeted elimination of ARGs, thereby establishing the technological underpinnings for the next generation of wastewater treatment processes [265].

### 4.3. Risk of Secondary Contamination of Treatment Byproducts

#### 4.3.1. Enrichment and Release of ARGs in Sludge

Sludge produced during wastewater treatment is an important reservoir of ARGs. The abundance of ARGs in dewatered sludge (1.80 × 10^5^–3.42 × 10^11^ copies/g) was significantly higher than that in influent water, and composting treatment reduced the abundance of ARGs by only 50% [64,266]. Agricultural sludge can contribute to the spread of ARGs through the food chain, e.g., sulfamethoxazole concentrations in carrots can reach up to 1.2 μg/g in agricultural fields where composted sludge is applied [267]. In addition, ARGs (e.g., blaCTX-M) in landfill leachate can contaminate groundwater and soil due to insufficient impermeability [268]. It has been shown that the accumulation of ARGs in sludge is closely related to HGT; in particular, the presence of integrons (intI1) and transposons significantly increases the risk of ARG spread [269].

Sludge is treated in various ways, but all are potentially risky: *sul1* and *sul2* genes are difficult to remove completely during composting, and a prolonged thermophilic period is required to reduce pathogens by optimising the carbon to nitrogen ratio (C/N = 30:1) [266]; natural zeolite additives reduce the incidence of HGT, but the practical results of large-scale composting are still inferior to those under laboratory conditions [270]. Some ARGs (e.g., sul1 and aadA) were instead enriched after anaerobic digestion (AD), and especially under mesophilic conditions, the abundance of ARGs was reduced by only 34% [228,234]. In addition, although high temperature digestion (55 °C) reduces some ARGs, the proliferation of thermophilic bacteria may promote the transfer of other drug resistance genes [271].

#### 4.3.2. Environmental Toxicity of Intermediate Degradation Products

The biotransformation of antibiotics can produce more toxic intermediates. For example, the degradation of amoxicillin to pyrazinic acid derivatives resulted in genotoxicity that was 1.8 times that of the parent compound [219], and tetracycline produced degradation products during photolysis that had a 40% lower EC50 for aquatic organisms [199]. The products formed by the amino oxidation of sulfamethoxazole (SMX) in membrane bioreactors (MBRs) lose their antimicrobial activity, but may disrupt the endocrine system [193]. Experiments have shown that SMZ degradation products significantly alter ovarian lipid and amino acid metabolism in zebrafish and increase gonadotropin-releasing hormone (GnRH) and sex hormone (e.g., oestradiol) levels, and that its degradation products may increase toxicity through endocrine disrupting effects [272].

The adsorption of antibiotics by conventional activated sludge (CAS) is the dominant removal mechanism, but degradation products (e.g., ciprofloxacin and tetracycline derivatives) pose a higher risk of toxicity [273]. In AGS systems, long sludge residence times (SRT) may increase the risk of antibiotic desorption, e.g., norfloxacin being re-released into the environment at a later stage [244]. The reversibility of the biosorption process of antibiotics such as tetracycline in microalgae technology also requires vigilance against secondary contamination [244,274]. In particular, photocatalytic degradation techniques can effectively reduce the formation of toxic intermediates, e.g., the TiO_2_/UV system mineralised ciprofloxacin degradation products by more than 90% [275].

## 5. Conclusions and Future Perspectives

This review systematically reveals the closed-loop threat chain formed by antibiotics and ARGs in the environment: from the widespread release of antibiotics and ARGs from medical, agricultural, and industrial sources, to their enrichment and spread in wastewater treatment systems driven by HGT (plasmid, integron, and outer membrane vesicle-mediated cross-species transmission are key pathways) [21], ultimately posing risks to human health through exposure pathways such as the food chain and aerosols. At the molecular level, bacteria, under the combined influence of various environmental stresses, develop multidrug resistance primarily through co-resistance mechanisms, including altered membrane permeability, the activation of efflux pumps, target site modifications, and enzymatic degradation. Additionally, biofilm resistance regulated by quorum sensing further enhances their resistance. HGT serves as the core mechanism for the dissemination of ARGs through conjugation, transformation, transduction, and outer membrane vesicles, relying on mobile genetic elements to facilitate cross-species transmission. Furthermore, core evidence indicates that biological treatment processes exhibit a significant “double-edged sword” effect: while activated sludge processes and membrane bioreactors (MBRs) can remove some antibiotic ARGs through extracellular polymeric substance (EPS) adsorption and QS regulation, their long sludge retention time (SRT) and high biomass density provide a proliferation environment for ARGs such as *sul1* and *tetM*, leading to a 1–2-fold increase in gene abundance in effluent compared to influent [12,64,223]. At the molecular level, bacteria develop multidrug resistance through altered membrane permeability, efflux pump activation, target site modification, and enzymatic degradation, while QS-regulated biofilm formation further enhances HGT efficiency [132,276]. Particularly concerning is the fundamental flaw in existing technologies for controlling secondary pollution: for example, the genetic toxicity of amoxicillin degradation products is 1.8 times higher than that of the parent compound [219], and antibiotics adsorbed by EPS in sludge can be re-released through desorption, exacerbating the risk of secondary pollution [191,244].

First, it is essential to address the gap in the toxicity assessment of metabolic intermediates. While existing studies have confirmed that the genetic toxicity of pyrazole acid derivatives generated from the degradation of amoxicillin is 1.8 times higher than that of the parent compound, and that the aquatic organism EC50 value of tetracycline photolysis products has decreased by 40% [199,219], such data only cover a limited number of compounds, and there is a severe lack of systematic toxicity assessments for complex transformation pathways such as hydroxylation and deamination. More urgently, the endocrine-disrupting and genetic mutation mechanisms of nanoscale transformation products (such as antibiotic–metal complexes) remain unclear, necessitating the development of standardized testing methods to establish a toxicity threshold database. This should be combined with environmental toxicology and computational toxicology models to quantitatively analyse their endocrine-disrupting and genetic mutation thresholds.

Secondly, it is necessary to overcome the technical bottleneck in HGT dynamic monitoring. Current strategies relying on bioinformatics (such as phylogenetic conflict analysis) may miss up to 98% of low-abundance transfer events due to the short read length limitation of metagenomic sequencing [277]. While emerging topological data analysis (such as “1-hole” feature identification) can distinguish between vertical inheritance and HGT, it cannot resolve the direction of transfer or quantify the stimulating effects of environmental stress (such as antibiotic pulse pollution) on transient transfer efficiency [278]. In the future, it will be necessary to integrate single-cell spatiotemporal tracking (such as FISH-nanopore combination) with artificial intelligence prediction models to construct an in situ monitoring platform capable of real-time capture of gene flow within biofilms.

Finally, there are still significant obstacles to be overcome in elucidating the molecular mechanisms: the key steps in transgenetic integration remain a “black box” (e.g., the mechanism by which Agrobacterium T-DNA hijacks the host DNA repair pathway in plants remains unclear [279], and the environmental signal thresholds for bacterial competence formation (e.g., the concentration of free DNA in soil pores that activates the com gene cluster) and the reprogramming patterns of environmental stress (e.g., heavy metals) on the transfer efficiency of the type IV secretion system (T4SS) have not yet been quantified [280]). The selective nature of vesicle-mediated transfer (vesiduction) (e.g., why marine microbial extracellular vesicles prefer to package chromosomal DNA rather than mobile elements) requires further investigation through cryo-electron microscopy to resolve the conformation of transmembrane channels (e.g., the dynamics of the ATPase in conjugation pili) [281], and synthetic biology should be used to construct genetic circuits to simulate the switching of phage host ranges under stress (e.g., the logic behind ΦSa1 acquiring the mecA gene) to elucidate its sorting basis.

On the basis of this scientific foundation, it is imperative that policymakers prioritize the establishment of a list of toxic degradation products and release limits for ARGs in sludge, incorporating these into the wastewater treatment regulatory framework. Concurrently, financial resources should be allocated to the development of an in situ monitoring network to warn of cross-border transmission of ARGs. Furthermore, efforts should be made to promote the integration and modelling of environmental–clinical resistance data to enable risk prediction from genes to population exposure.

## Figures and Tables

**Figure 1 microorganisms-13-02113-f001:**
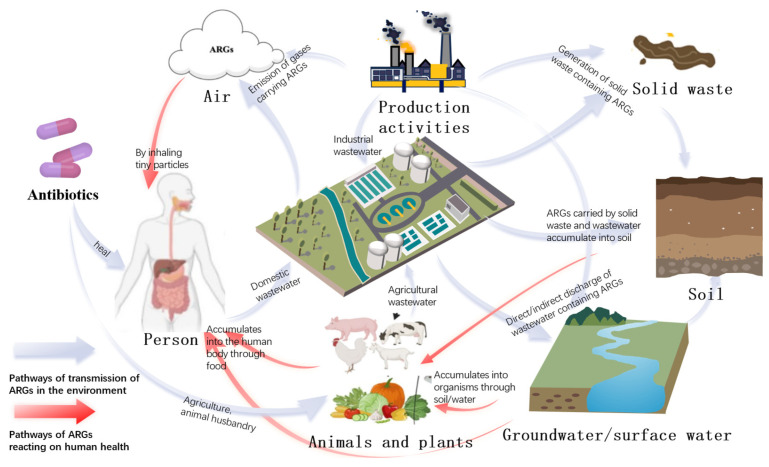
Life cycle of antibiotic resistance genes in the environment.

**Figure 2 microorganisms-13-02113-f002:**
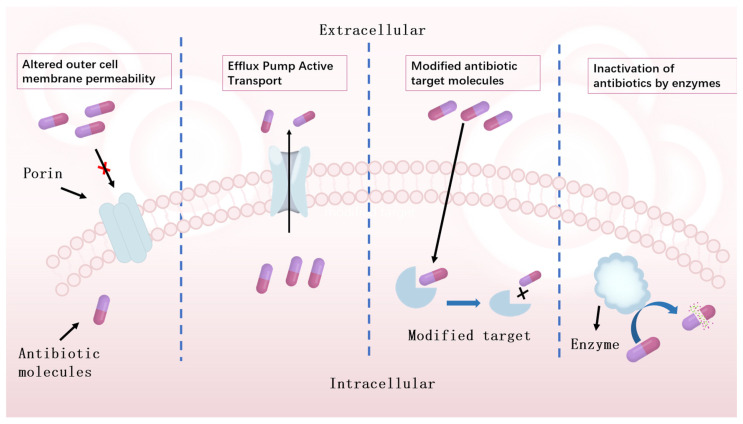
Mechanisms of inherent bacterial resistance.

**Figure 3 microorganisms-13-02113-f003:**
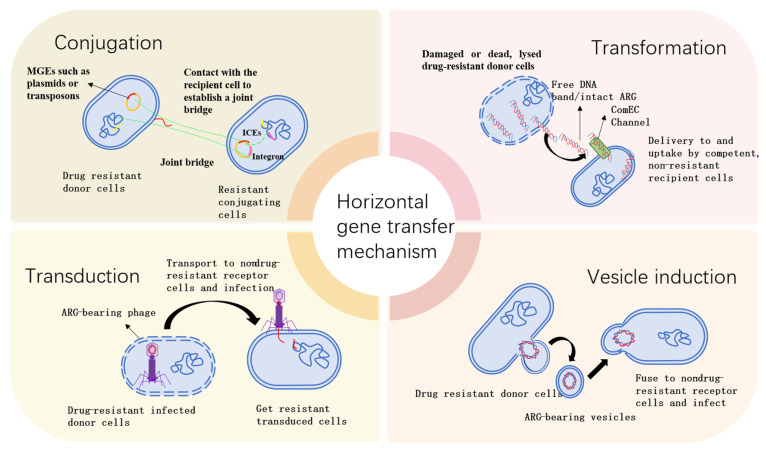
Bacterial acquired resistance—horizontal gene transfer mechanisms.

## Data Availability

No new data were created or analysed in this study.

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
