# Peer review of "The Environmental Lifecycle of Antibiotics and Resistance Genes: Transmission Mechanisms, Challenges, and Control Strategies"

_microorganisms, 2025, doi:10.3390/microorganisms13092113_

Round 1

Reviewer 1 Report

Comments and Suggestions for Authors

General Comments:

  1. After careful and critical reading of this systematic review, it was easy to understand that the authors (Li, et al.) are summarizing the available information and highlighting the full life cycle of antimicrobial compounds and microorganism-carried antibiotic resistance genes in the environment, from widespread release from medical, agricultural and industrial sources.
  2. The document appears to be well-organized and follows the scientific conventions of systematic review articles.
  3. The authors provide an introduction with sufficient background, but a good overview of the topics and their importance of helping support decision-making on antibiotics usage.
  4. Since this is a systematic review, the cited reference list appears to be extremely abundant and relevant to the topic, providing additional information for readers who want to learn more details. Sixty-nine of 216 references are from the last 5 years, 95/216 are from 6 to 10 years ago and 52 of 216 are older than 10 years. Six hundred and fifty-nine (659) sentences are referenced in the manuscript. Some in blue and some in black. Which is the difference? Please, correct the format of reference number 81.
  5. The manuscript structure appears to be appropriate for the review design and topic being addressed and provides a comprehensive investigation of the scenario of antibiotics and antibiotic resistance genes worldwide in the last decades, regarding the dynamic fate in wastewater treatment processes, and ultimately rethreatening human health through the food chain and aerosols.
  6. The written summarized contents are presented clearly, and I assume they are well-supported by the data properly organized in the form of Tables (1 and 2) and Figures (1 and 2), of which, I suggest organizing better the categories in the first column. Avoid repeating and try to group.
  7. Overall, the updated knowledge displayed in the study is well-described and provides sufficient information for readers to understand the topics. This will be useful to the scientific and environmental community in terms of comparability and reproducibility.
  8. However, it is highly recommended to improve the quality of the manuscript by including an overview of the antimicrobial compounds of environmental impact and importance, as well as their classification.
  9. Please, re-write the paragraph corresponding to section 2.1 Sources and Emission of Antibiotics. Lines 79 to 84. Authors begin talking about antibiotics, then shift to microorganisms in the same idea and finalize referencing Figure 1., which has a totally wrong legend (Figure 1. Life cycle of antibiotic resistance genes in the environment). It is about antimicrobial compounds, microorganisms or genes?
  10. Similarly, it is important to clarify the hierarchy of the taxonomic groups addressed in the manuscript. For example: Bacterial Family, Genus and/or Species discussed between lines 358 and 371 in the manuscript are really confusing.
  11. Despite I am not qualified to assess the quality of English in this manuscript, overall, the quality of English language in the document is appropriate for a scientific research article.
  12. The authors should discuss more deeply the strengths and limitations of their study and suggest that this article could potentially serve as basis for future health policies regarding antibiotics and antibiotic resistance genes.
  13. Overall, the document appears to be a well-designed and well-executed scientific review, and it is likely to be of interest to readers who are interested in the topic.
  14. The conclusions drawn by the authors are supported by the knowledge displayed in this systematic review. This study summarizes important findings to be reminded by the scientific community, helping support future decision-making on antibiotics and antibiotic resistance genes.

Minor comments and suggestions:

  1. The nomenclature of genes and proteins are mixed. Please systematically seek and unify all the names in the tables and the manuscript: genes must be in italics and proteins in non-italics with the first position in a capital letter.
  2. Please, provide better usage for the abbreviations throughout the manuscript. An abbreviation must be introduced and then further used.

Example 1:

Antibiotic Resistance Genes is displayed 3x in the manuscript.

Antibiotic Resistance Genes (ARGs) is displayed 2x in the manuscript.

ARGs are displayed 152x in the manuscript.

Example 2:

Vertical Gene Transfer is displayed 3x in the manuscript.

Vertical Gene Transfer (VGTs) is displayed 2x in the manuscript.

VGTs are not displayed at all in the manuscript.

Example 3:

Mobile Genetic Elements are displayed 11x in the manuscript.

Mobile Genetic Elements (MGEs) are displayed 5x in the manuscript.

MGEs are displayed only once in the manuscript.

  1. Please, introduce spaces to separate some words at line 49 [(ARGs).ARBs], line 702 [removal.CAS] and erase the space in lane 704 [( SBR)].

Author Response

Reviewer #1

Response to the reviewers’ comments

First, we would like to thank the reviewers and the editor for the positive and constructive comments and suggestions,we have carefully checked the manuscript and the modifications were noted in the revised manuscript with changes marked (in red).

  1. After careful and critical reading of this systematic review, it was easy to understand that the authors (Li, et al.) are summarizing the available information and highlighting the full life cycle of antimicrobial compounds and microorganism-carried antibiotic resistance genes in the environment, from widespread release from medical, agricultural and industrial sources.

Response:Thank you for your careful review of our manuscript . We greatly appreciate your positive comment recognizing that our systematic review effectively summarizes the available information and highlights the full life cycle of antimicrobial compounds and microorganism-carried antibiotic resistance genes in the environment, tracing their release from diverse medical, agricultural, and industrial sources.We are pleased that the core focus and contribution of our work were clearly communicated.

  1. The document appears to be well-organized and follows the scientific conventions of systematic review articles.

Response:Thank you for your positive feedback on the organization and scientific rigor of our systematic review manuscript . We are pleased to know that the structure aligns with conventional standards for such articles.

  1. The authors provide an introduction with sufficient background, but a good overview of the topics and their importance of helping support decision-making on antibiotics usage.

Response:Thank you for recognizing the adequacy of the background in our introduction and the relevance of our review for supporting decision-making on antibiotic usage. We deliberately structured the introduction to bridge scientific evidence with practical applications, and we are gratified that this intent was clear.

  1. Since this is a systematic review, the cited reference list appears to be extremely abundant and relevant to the topic, providing additional information for readers who want to learn more details. Sixty-nine of 216 references are from the last 5 years, 95/216 are from 6 to 10 years ago and 52 of 216 are older than 10 years. Six hundred and fifty-nine (659) sentences are referenced in the manuscript. Some in blue and some in black. Which is the difference? Please, correct the format of reference number 81.

Response:Thank you. There's no difference, I've standardized the formatting across the board.

  1. The manuscript structure appears to be appropriate for the review design and topic being addressed and provides a comprehensive investigation of the scenario of antibiotics and antibiotic resistance genes worldwide in the last decades, regarding the dynamic fate in wastewater treatment processes, and ultimately rethreatening human health through the food chain and aerosols.

Response:Thank you for your insightful assessment of our manuscript. We are particularly gratified by your recognition that:The structure aligns with the review's objectives,The analysis comprehensively traces the global pathways of antibiotics and resistance genes — from wastewater treatment dynamics to their re-entry into human systems via food chains and aerosols.Highlighting this interconnected threat to public health was indeed a core motivation for our work. We appreciate your validation of its significance.

  1. The written summarized contents are presented clearly, and I assume they are well-supported by the data properly organized in the form of Tables (1 and 2) and Figures (1 and 2), of which, I suggest organizing better the categories in the first column. Avoid repeating and try to group.

Response:Thanks, the Table 1 has been re-combined and categorized in Form 1.

Table 1. Abundance of ARGs in different sources of WWTP.

Source

ARGs

Content (copies/ g dry weight)

References

Municipal Wastewater Treatment Plants

tet(A), tet(B), tet(E), tet(G), tet(H), tet(S), tet(T), tet(X), sul1, sul2, qnrB, and erm(C)

(1.5±2.3)×109 - (2.2±2.8)×1011

[20]

Municipal wastewater treatment plant effluents

tetA, tetC, tetG, tetM, tetO, tetW, tetX, sul1, sul2

3.6×101(teW) to 5.4×106

(tetX) copies mL-I

6.4×1012(tetW)to 1.7×1018

(sull) copics d-1

[21]

Hospital Wastewaters

blaOXA-48CTX-MblaIMP blaTEM

5.36×1011 - 1.90×1012

[22]

sul1, blaSHV, catA1 ,aacC2, tetA;

1.94×101,4.39×10-3,6.83×10-5,5.67×10-3,3.46×10-3;copies/16S rRNA gene copies

[23]

sul1, sul2, sul3, tetQ;

1.79×101~6.67×101,7.33×10-2~3.38×101,9.22×10-2~5.9×101,2.8×101~7.47×101;copies/16S rRNA gene copies

[24]

Livestock wastewater

tetL, strB, sul2, tetG, ermB, sul1, tetX,and cmlA

tetL(1.36~0.39)copies/ml/16Rrna,−−strB(0.82~0.52),sul2(0.96~0.64),tetG−−(1.81~0.67),ermB(1.17~0.71), sul1−− ( 1.51 ~ 0.93), tetX ( 1.17 ~ 0.94), and cmlA −− ( 1.73 ~ 1.14)

[25]

tetX, ermF, ermB, mefA, tetM, sul2

2.43×1011- 5.69×1010

copies/mL

[26]

sul1, sul2, tetM;

3.84×101, 1.62×101, 2.33×101;copies/16S rRNA gene copies

[27]

tetC, tetO

7.3×103, 1.7×101;copies/16S rRNA gene copies

[21]

Pharmaceutical industries wastewater

etA, tetC, tetG, tetL, tetM, tetO,

1.4×101, 3.2×102, 5.1×102, 6.1×102, 1.1×102, 1.0×100, 1.8×100, 1.6×101, 3.7×103;copies/16S rRNA gene copies

[28]

sul1, sul2, tetA, qacE, qacED1;

101 to 102;copies/16S rRNA gene copies

[29]

Reclaimed water irrigation

tetG, tetW, sulI, sulII, intI1

Highest abundance of sulII and intI1, the abundances − 71 were 8.43×10 copies g − 71 dry soil, 7.62×10 copies g dry soil

[30]

Drinking water treatment plants

sul I, sul II, tet(C), tet(G), tet(X), tet(A), tet(B), tet(O), tet(M), and tet(W)

Total concentrations of ARGs belonging to either the sulfonamide or tetracycline resistance gene class wereabove 105 copies mL-1

[31]

river sediments

TEM,sul1,sul2

1.09 × 10−1

1.06 × 10−1copy of 16S-rRNA gene

[32]

Table 2. Disposal efficiency of ARGs in different biological treatments.

Processing techniques

Operating conditions

ARGs kind

Removal effect

References

Activated sludge treatment

tetO, tetW

3 logs

[202]

ermB,tetW,sul2

1.29–2.45 log (ermB), 1.13–1.62 log (tetW), 0.26–0.53 log (sul2)

[63]

CASS

tetAtetOtetWsulIsulIIblaCTX-M

>2.60 ± 0.015 log (tetO); >2.66 ± 0.023 log (tetW)

[203]

A/O

tet,erm,sul,qnr,bla

16.90% (total ARGs), 64.50% (tet), 92.00% (erm)

[200]

A/A/O

tet,erm,sul,qnr,bla

56.00% (tet), 70.40%–87.00% (erm)

[200]

sulI, sulII, tetO, tetW, tetQ

1.69 logs, 1.4 4 logs, 2.31 logs, 2.13 logs, 2.5 logs

[204]

Membrane bioreactor

sulII, tetO, tetW

2.57 logs, 7.06 logs, 6 logs

[205]

anMBR

blandM1, blaCTXM15, and blaOXA48

2.76–3.84 logs

[206]

A/O-MBR

sulI, sulII, tetC, tetX, ereA, and int1

0.5–5.6 logs

[11]

Aerobic granular sludge

tetW,sul2,sul1,intI1,ermB

2.02 log (tetW), 1.43 log (sul2), 0.77 log (sul1), 0.55 log (intI1), 0.08 log (ermB)

[63]

Anaerobic digestion

40 °C, 56 °C, 60 °C, and 63 °C.

tetW, tetX, qnrA, and intI1.

Decreased ARGs except qnrA by 89 96 % and ~ 99 % at 40 °C and other temperatures, and decreased qnrA by 99 % at 40, 60, and 63 °C.

[207]

MAD

35 °C, sludge retention time (SRT) 20 d.

sulI, sulII, tetA, tetO, tetX, bla , and TEM bla .

Decreased extracellular ARGs by 0.11 1.22 logs.

[208]

TAD

55 °C, SRT 20d.

sulI, sulII, tetA, tetO, tetX, bla , and TEM blaSHV.

Decreased extracellular ARGs by 0.33 1.46 logs.

[208]

composting

kitchen waste

etA, tetB, tetC, tetG, tetM, tetO, tetQ, tetW, tetX, sul1, sul2, sul3, and dfrA7, qnrB, qnrS, acc(6)-Ibcr, ermB, ermF, ermQ, ermX, mefA,

total ARGs: 99.68%–99.98% (tetracyclines: >99%; sulfonamides: 5.35%–8534.69%; quinolones: 837.30%–99.29%; macrolides: 4425.46%–98.14%)

[209]

cattle manure

ermB, ermF, ermQ, ermX, sul1, sul2, sulA, tetA, tetB, tetC, tetE, tetG, tetK, tetM, tetO, tetQ, tetW, tetX

total ARGs: 52.69%

[210]

sewage sludge

ermB, ermC, sul1, sul2, tetC, tetG, tetO

ermB, ermC, sul1, and tetC: 25.7%, 42.4%, 69.4%, and 44.6%, respectively

[211]

CW-surface flow

Capacity:600m/d HLR:350-450mm/d

HRT:6h

sulI, sulII, sulIII, tetA, tetB, tetC, tetE, tetH, tetM, tetO, tetW, qnrB, qnrS, and qepA

77.8% in summer, 59.5% in winter

[212]

CW-horizontal subsurface flow

Capacity: 500 m /d

intl1, sulI, sulII, dfrA, aac6, tetO, qnrA, blaNMD1, blaKPC, blaCTX, and ermB

145.6%–98.9%

[213]

CW-vertical subsurface flow

HLR: 5.1 cm/d

tet genes and intI1

33.2%–99.1%

[214]

Constructed wetland

sulI, sulII, tetO, tetW and tetQ

1.5 logs, 0.48 log, 2.1 logs, 1.5 logs, 2.1 logs

[204]

microalgae

bla-Tem,ermB

0.56logs,1.75logs

[215]

sul1, tetQ, blaKPC, and intl1

1.2-4.9logs,2.7-6.3logs,0-1.5logs,1.2-4.8logs

[216]

  1. Overall, the updated knowledge displayed in the study is well-described and provides sufficient information for readers to understand the topics. This will be useful to the scientific and environmental community in terms of comparability and reproducibility.

Response:Thank you for acknowledging the updated knowledge coverage and our manuscript's utility for scientific reproducibility/comparability. We are encouraged that these features align with community needs.

  1. However, it is highly recommended to improve the quality of the manuscript by including an overview of the antimicrobial compounds of environmental impact and importance, as well as their classification.

Response:Thanks. In response to this issue, we have added a new subsection 2.1.4, ''Major Antibiotic Families and Antibiotic Resistance Gene (ARG) Subtypes in the Environment.''Line192-233

  1. Please, re-write the paragraph corresponding to section 2.1 Sources and Emission of Antibiotics. Lines 79 to 84. Authors begin talking about antibiotics, then shift to microorganisms in the same idea and finalize referencing Figure 1., which has a totally wrong legend (Figure 1. Life cycle of antibiotic resistance genes in the environment). It is about antimicrobial compounds, microorganisms or genes?

Response:Thank Lines 79-84 in section 2.1 of the article have been revised, and the arrows in the figure indicate the cycling path of ARGs through the environment.

''The extensive utilization of antibiotics in healthcare, agriculture, aquaculture, and livestock farming has been demonstrated to directly induce the evolution of microbial resistance through the prevention and treatment of bacterial diseases. [19, 20]These pharmaceuticals and their metabolites are continuously introduced into environmental media through multiple pathways, including hospital wastewater discharge, human and animal excreta, the release of products containing antimicrobial active ingredients, and contaminated feed and the food chain. Once in the environment, antibiotics act as a strong selective pressure, driving the enrichment and amplification of antibiotic resistance genes (ARGs) within bacterial communities. Furthermore, mobile genetic elements (such as plasmids and integrons) facilitate the horizontal transfer and recombination of ARGs in various habitats, including soil, water bodies, and biofilms. This phenomenon contributes to the persistent spread and accumulation of resistance within environmental microbial networks.''

''Figure 1 depicts the cycle of ARGs in the environment. The blue lines represent the transmission path of ARGs from humans to the environment, while the red lines represent the cycle of ARGs back to humans.''

  1. Similarly, it is important to clarify the hierarchy of the taxonomic groups addressed in the manuscript. For example: Bacterial Family, Genus and/or Species discussed between lines 358 and 371 in the manuscript are really confusing.

Response:Thanks. This section has been deleted and replaced with:''In the context of environmental pollution, pharmaceutical industrial wastewater has been identified as a significant catalyst. For instance, Klebsiella pneumoniae carrying the blaNDM-1 and intI1 integrons was detected in hospital wastewater treatment plants in India, with antibiotic resistance gene abundance four times higher than in clinical strains. This phenomenon is primarily attributable to synergistic selection pressure from residual carbapenems (>200 μg/L) and heavy metals present in the wastewater [124]. The enzymatic mechanisms present in clinical strains are often the result of plasmid-mediated horizontal gene transfer within healthcare facilities [97].''

  1. Despite I am not qualified to assess the quality of English in this manuscript, overall, the quality of English language in the document is appropriate for a scientific research article.

Response:Thank you for your general assessment regarding the English quality of our manuscript. We appreciate your confirmation that the language is appropriate for scientific communication. Should any specific linguistic refinements be identified during subsequent review stages, we will promptly address them.

  1. The authors should discuss more deeply the strengths and limitations of their study and suggest that this article could potentially serve as basis for future health policies regarding antibiotics and antibiotic resistance genes.

Response:Thanks. In response to this question, we have added a discussion of the remaining issues in the current research in the conclusion and outlook section at the end.

''Despite the elucidation of the three classic mechanisms (conjugation, transformation, and transduction), the molecular intricacies of transgenetic integration persist as a "black box." For instance, the process of Agrobacterium T-DNA integration in plants is contingent upon the host DNA repair pathways (e.g., non-homologous end joining). However, the mechanism through which it is hijacked by pathogens remains to be elucidated [307]. The environmental signal thresholds for bacterial competence formation are also not yet known. It is not yet known what concentration of free DNA in soil pores is required to activate the com gene cluster. The present study aims to investigate the impact of environmental stresses, such as heavy metals, on the transfer efficiency of the type IV secretion system (T4SS). [308]. The emerging vesicle-mediated transfer (vesiduction) mechanism is also the subject of ongoing research regarding selectivity. Marine microbial extracellular vesicles (EVs) have been observed to preferentially package chromosomal DNA rather than mobile elements. However, the molecular basis for this selective sorting remains unclear [309]. Future research should integrate cryo-electron microscopy to resolve the conformational dynamics of transmembrane channels (e.g., the ATPase activity of fimbriae), and utilize synthetic biology to construct genetic circuits simulating the switching mechanisms of phage host range under environmental stress (e.g., the regulatory logic of ΦSa1 acquiring the mecA gene in Staphylococcus aureus) [310].''

  1. Overall, the document appears to be a well-designed and well-executed scientific review, and it is likely to be of interest to readers who are interested in the topic.

Response:Thank you for your comprehensive endorsement of the manuscript's scholarly quality and anticipated reader interest.

  1. The conclusions drawn by the authors are supported by the knowledge displayed in this systematic review. This study summarizes important findings to be reminded by the scientific community, helping support future decision-making on antibiotics and antibiotic resistance genes.

Response:Thank you very much for your positive feedback and for handling our manuscript. We are very pleased to hear that you found the conclusions of our systematic review to be well-supported and that you consider the summarized findings important for reminding the scientific community and supporting future decision-making regarding antibiotics and antibiotic resistance genes.

Minor comments and suggestions:

  1. The nomenclature of genes and proteins are mixed. Please systematically seek and unify all the names in the tables and the manuscript: genes must be in italics and proteins in non-italics with the first position in a capital letter.

Response: Thanks. We have standardized the names of bacteria, genes, and proteins in the article one by one.

2.Please, provide better usage for the abbreviations throughout the manuscript. An abbreviation must be introduced and then further used.

Example 1:

Antibiotic Resistance Genes is displayed 3x in the manuscript.

Antibiotic Resistance Genes (ARGs) is displayed 2x in the manuscript.

ARGs are displayed 152x in the manuscript.

Example 2:

Vertical Gene Transfer is displayed 3x in the manuscript.

Vertical Gene Transfer (VGTs) is displayed 2x in the manuscript.

VGTs are not displayed at all in the manuscript.

Example 3:

Mobile Genetic Elements are displayed 11x in the manuscript.

Mobile Genetic Elements (MGEs) are displayed 5x in the manuscript.

MGEs are displayed only once in the manuscript.

Response: Thanks. We have revised each abbreviation in the article one by one.

3 Please, introduce spaces to separate some words at line 49 [(ARGs).ARBs], line 702 [removal.CAS] and erase the space in lane 704 [( SBR)].

Response: Thanks.we have completed the revisions.The first instance shouldbereplaced with other content.''removal . CAS''''(SBR)''

Reviewer 2 Report

Comments and Suggestions for Authors

Please take care colors off fig1. Cursive letters as:  Klebsiella pneumoniae lines 280, also  305, 306, 326,342, 364, 410, 452, 453, 536, 540, 541, 775,777, 779, 780

1-producing Klebsiella pneumoniae) in hospital wastewater enter wastewater treatment 280

In pathogens such as Klebsiella pneumoniae, Enterobacter aerogenes, Neisseria gon- 305 orrhoeae and Pseudomonas aeruginosa, mutations in specific pore proteins such as 306

For example, the AcrABZ-TolC system in E. coli consists of four different func- 326

Mycobacterium tuberculosis confer resistance to strepto- 342

Line 352    eliminate .123124

Escherichia coli 364 and Streptomyces fowleri

Staphylococcus aureus, 410

Line 380and  400  Needs reference        

 binding molecules that neutralise or bind AMP. 380

bile genetic elements (MGEs) such as plasmids, transposons, integrons and phages. 400

line 402   capital letter     3.2.1. conjugation: must be Congugation

Comments on the Quality of English Language

Please take care colors off fig1. Cursive letters as:  Klebsiella pneumoniae lines 280, also  305, 306, 326,342, 364, 410, 452, 453, 536, 540, 541, 775,777, 779, 780

1-producing Klebsiella pneumoniae) in hospital wastewater enter wastewater treatment 280

In pathogens such as Klebsiella pneumoniae, Enterobacter aerogenes, Neisseria gon- 305 orrhoeae and Pseudomonas aeruginosa, mutations in specific pore proteins such as 306

For example, the AcrABZ-TolC system in E. coli consists of four different func- 326

Mycobacterium tuberculosis confer resistance to strepto- 342

Line 352    eliminate .123124

Escherichia coli 364 and Streptomyces fowleri

Staphylococcus aureus, 410

Line 380and  400  Needs reference     

 binding molecules that neutralise or bind AMP. 380

bile genetic elements (MGEs) such as plasmids, transposons, integrons and phages. 400

line 402   capital letter     3.2.1. conjugation: must be Congugation

Author Response

Reviewer #2

First, we would like to thank the reviewers and the editor for the positive and constructive comments and suggestions,we have carefully checked the manuscript and the modifications were noted in the revised manuscript with changes marked (in red).

  1. Antibiotic residues in the environment exert continuous selection pressure, thereby 48 inducing the proliferation of drug-resistant bacteria (ARBs) and antibiotic resistance 49 genes (ARGs).ARBs maintain population stability through vertical gene transfer (VGT), 50 whereas horizontal gene transfer (HGT) facilitates the dissemination of ARGs across 51 species by means of mobile genetic elements (MGEs), such as plasmids and transposons, 52 and even the formation of multi-drug resistant bacteria (MDRB) [7-9].

Response:Tanks,I have revised the original sentence.It is as follows:"Antibiotics remaining in the environment exert continuous selective pressure on microorganisms, thereby inducing the production of more antibiotic-resistant bacteria (ARBs). The resistance genes of ARBs can be acquired through vertical gene transfer (VGT), but they are mainly acquired through horizontal gene transfer (HGT) within species via mobile genetic elements (MGEs) (such as plasmids and transposons), thereby potentially forming multidrug-resistant bacteria (MDRB)."

  1. wastewater treatment plants (WWTPs) 56 represent pivotal obstacles to antibiotic resistance genes (ARGs),

Response:Tanks,I have revised the original sentence.It is as follows:"Wastewater treatment plants are key points for the accumulation of resistance genes."

  1. processes (e.g., activated sludge, membrane bioreactors) have the po- 58 tential to function as 'amplifiers' of ARGs.

Response:Tanks,I have revised the original sentence.It is as follows:"processes (e.g., activated sludge, membrane bioreactors) carry the risk of enriching ARGs."

  1. Furthermore, the molecular mechanisms by which 65 the group sensing (QS) system regulates exopolysaccharide (EPS) secretion and biofilm 66 formation have not been fully elucidated in environmental resistance studies [15] . 67

Response:Tanks,I have revised the original sentence.It is as follows:"In addition, the resistance mechanisms of biofilms regulated by the quorum sensing (QS) system in the environment have not yet been fully elucidated."

  1. propose multidisciplinary synergistic management strat- 75 egies, so as to provide a scientific basis for curbing the global drug resistance crisis. 76

Response:Tanks,I have revised the original sentence.It is as follows:"propose multidisciplinary collaborative management strategies to support efforts to curb the global antibiotic resistance crisis."

  1. o animals through the use of antibiotic-containing food and feed 83 (Figure 1). 84

Response:Tanks,I have revised this paragraph.It is as follows:"The extensive utilization of antibiotics in healthcare, agriculture, aquaculture, and livestock farming has been demonstrated to directly induce the evolution of microbial resistance through the prevention and treatment of bacterial diseases. [18, 19]These pharmaceuticals and their metabolites are continuously introduced into environmental media through multiple pathways, including hospital wastewater discharge, human and animal excreta, the release of products containing antimicrobial active ingredients, and contaminated feed and the food chain. Once in the environment, antibiotics act as a strong selective pressure, driving the enrichment and amplification of antibiotic resistance genes (ARGs) within bacterial communities. Furthermore, mobile genetic elements (such as plasmids and integrons) facilitate the horizontal transfer and recombination of ARGs in various habitats, including soil, water bodies, and biofilms. This phenomenon contributes to the persistent spread and accumulation of resistance within environmental microbial networks."

  1. As a result of their function as a focal point for 94 wastewater treatment, WWTPs have, regreĴably, also become a breeding ground for the 95 spread of ARGs [20]

Response:Tanks,I have revised the original sentence.It is as follows:"Unfortunately, wastewater treatment plants, as the center of wastewater treatment, have also become breeding grounds for the spread of ARGs."

  1. Please take care colors off fig1 Figure 1, the cycling pathways of ARGs in the environment are represented, 89 with the yellow line signifying the impact of human activities and the green line denoting 90 the natural cycling process.

Response:Tanks, we have changed the colors explained in the text."Figure 1 depicts the cycle of ARGs in the environment. The blue lines represent the transmission path of ARGs from humans to the environment, while the red lines represent the cycle of ARGs back to humans."

  1. For instance, 109 the concentration of ARGs in residential wastewater can range from 2.28 × 1011 to 3.41 × 110 1011 copies/mL, which is significantly higher than that of hospital wastewater, which 111 ranges from 4.62 × 107 to 5.70 × 1011 copies/mL [35].

Response:Tanks,I have revised the original sentence.It is as follows:"For example, the concentration range of ARGs in the wastewater of a certain resident was 2.28 × 10¹¹ to 5.70 × 10¹¹ copies/mL, and this abnormal phenomenon may be related to the abuse of antibiotics in the community[35]."

  1. the mixing of hospital wastewater with domestic wastewater in wastewater treat- 125 ment plants may accelerate the cross-transmission of ARGs through processes such as bio- 126 film formation and sludge adsorption. This ‘low-flow-high-toxicity’ characteristic makes 127

Response:Tanks,I have revised the original sentence.It is as follows:" Notably, the mixing hospital wastewater with domestic wastewater in wastewater treatment plants may cause cross-contamination of ARGs. This type of “low flow-high toxicity” mixed wastewater has also become a key hotspot for the spread of drug resistance."

  1. (e.g., tetracyclines, sulphona- 134 mides) are frequently incorporated into animal feed with the intention of promoting 135 growth and preventing or controlling diseases

Response:Tanks,I have revised the original sentence.It is as follows:"(e.g , tetracycline and sulfonamide) are often found in animal feed."

  1. These findings suggest that the role of plants as 153 carriers of ARGs should not be ignored [45]

Response:Tanks,I have revised the original sentence.It is as follows:"These findings indicate that the risk of plants as carriers of ARGs should not be overlooked."

  1. The 'heavy metal-antibiotic' syner- 159 gistic effect significantly accelerated the spread of ARGs in the soil microbiota by facilitat- 160 ing the translocation of mobile genetic elements (MGEs) by conjugation. 161

Response:Tanks,I have revised the original sentence.It is as follows:"The synergistic effect of “heavy metals-antibiotics” promotes the transfer of MGEs through conjugation, which can significantly accelerate the spread of ARGs in soil microbial communities."

  1. the term co-regulation was proposed, signifying a series of transcriptional and 188 translational reactions that form a coordinated response to metals or antibiotics following 189 exposure to stress. These mechanisms result in the continuous screening and enrichment 190 of ARGs in bacteria, which ultimately leads to increased levels of resistance [56] . 191

Response:Tanks,I have revised the original sentence.It is as follows:"Another mechanism is co-regulation, which refers to a series of transcriptional and translational behaviors in response to the combined stress of exposure to heavy metals and antibiotics. These mechanisms lead to the continuous screening and enrichment of ARGs in bacteria, ultimately resulting in increased levels of resistance."

  1. Furthermore, 201 the discharge of treated wastewater has been shown to result in a more than twofold in- 202 crease in ARGs abundance downstream of a receiving water body compared to upstream 203

Response:Tanks,I have revised the original sentence.It is as follows:"Furthermore, compared with upstream, the discharged wastewater caused the abundance of ARGs in downstream water bodies to increase more than twofold."

  1. . It has been demonstrated that, following re-release through 213 hydraulic flushing, the propagation of resistance within the water column is intensified. 214

Response:Tanks,I have revised the original sentence.It is as follows:"It has been demonstrated that these captured ARGs are re-released into the water body after hydraulic flushing, exacerbating the spread of drug resistance in the water body."

  1. aerosols carrying blaCTX-M-15 genes can migrate to communities 237 up to 10 km away and directly threaten human health through lung colonisation [60]. It is 238 worth noting that the survival of ARGs in aerosols is regulated by UV and humidity, with 239 a greater dispersal radius in arid regions. 240

Response:Tanks,I have revised the original sentence.It is as follows:"aerosols carrying blaCTX-M-15 genes can migrate to communities 237 up to 10 km away and directly threaten human health through lung colonisation [60].It is worth noting that the survival capacity of ARGs in aerosols is affected by ultraviolet rays and humidity, and it will have a greater diffusion radius in arid regions."

  1. water sources (sewage, surface water), suggesting that water treatment processes cannot 247 completely block ARG transmission [62, 63].

Response:Tanks,I have revised the original sentence.It is as follows:"However, its diversity is highly consistent with sewage, indicating that the drinking water treatment process cannot completely block the spread of ARGs."

  1. For example, aerosol carriers of Acinetobacter baumannii and Staphylococcus au- 271 reus can detect blaNDM-1 and mecA gene amplification in lung tissue within 48 h of in- 272 halation, triggering drug-resistant infections [60, 69].

Response:Tanks,I have revised the original sentence.It is as follows:"For example, Acinetobacter baumannii and Staphylococcus aureus carrying the blaNDM-1 and mecA genes were detected to have amplified within 48 hours after being inhaled into lung tissue via aerosols, causing drug-resistant infections."

  1. Cursive letters as: Klebsiella pneumoniae lines 280, also 305, 306, 326,342, 364, 410, 452, 453, 536, 540, 541, 775,777, 779, 780,

Response:Thank you, the changes have been made.

  1. 1-producing Klebsiella pneumoniae) in hospital wastewater enter wastewater treatment 280

Response:Thank you, the changes have been made.

  1. In pathogens such as Klebsiella pneumoniae, Enterobacter aerogenes, Neisseria gon- 305 orrhoeae and Pseudomonas aeruginosa, mutations in specific pore proteins such as 306

Response:Thank you, the changes have been made.

  1. For example, the AcrABZ-TolC system in E. coli consists of four different func- 326

Response:Thank you, the changes have been made.

  1. Mycobacterium tuberculosis confer resistance to strepto- 342

Response:Thank you, the changes have been made.

  1. Line 352 eliminate .123124

Response:Thank you, the changes have been made.

  1. Escherichia coli 364 and Streptomyces fowleri

Response:Thank you, the changes have been made.

  1. Staphylococcus aureus, 410

Response:Thank you, the changes have been made.

  1. Line 380and 400 Needs reference

Response:Thank you, the changes have been made.

  1. binding molecules that neutralise or bind AMP. 380

Response:Thank you, the changes have been made.

  1. bile genetic elements (MGEs) such as plasmids, transposons, integrons and phages. 400

Response:Thank you, the changes have been made.

  1. line 402 capital letter 3.2.1. conjugation: must be Congugation

Response:Thank you, the changes have been made.

  1. line Strep- 452 tococcus pneumoniae, Bacillus subtilis and Vibrio cholerae, are known to possess natural 453 transformation competence. E. coli is the preferred model for transformation studies due 454

Response:Thank you, the changes have been made.

  1. Line 467 Needs reference via the DprA protein. 467

Response:Thanks, this part has been deleted after modification.

  1. Line 536 Escherichia coli is 85% efficient in effluxing 536

Response:Thank you, the changes have been made.

  1. Line 540, 541 cursive letter Asper- 541 gillus chimaera

Response:Thank you, the changes have been made.

  1. Line 671 Need reference the risk of ARG regeneration. 671

Response:Thank you, the changes have been made.

  1. Line 775 Cursive letter Pseudomonas aeru- 775 ginosa

Response:Thank you, the changes have been made.

  1. Line 77 cursive letter Chlorella vulgaris was significantly affected by the initial con- 777

Response:Thank you, the changes have been made.

  1.  Line 779 cursive letter Chlorella proteolytica

Response:Thank you, the changes have been made.。

  1. Micro- 779-780 cystis aeruginosa [205 780

Response:Thank you, the changes have been made.

Reviewer 3 Report

Comments and Suggestions for Authors

In the manuscript "The Environmental Lifecycle of Antibiotics and Resistance Genes: Transmission Mechanisms, Challenges, and Control Strategies", the Authors aim to present the fate of antibiotic resistance genes and bacteria after they leave the places where antibiotic treatment is performed, and transfer to the environment or treatment plants. 

The topic is interesting, and indeed deserves a thorough review, however, i have some major concerns with this manuscript:

  • The writing style is inconsistent. Some parts of the manuscript are too simple for a review article; e.g. in chapters 3.1-3.2, very basic information is presented, which does not fit well with the topic. It also does not cover all the necessary pathways that bacteria can exchange DNA. Other chapters, on the contrary, are heavy with technological terms, presenting water treatment plant operations and tools without clear explanations.
  • Not all information presented in the text has appropriate references; some references do not contain information that the Authors claim.
  • The information in figures and tables is quite confusing - figure 1 does not have yellow or green lines referenced in the text, in figures 2-3 the arrows are not of the correct orientation, the tables present a lot of data, however, it is not of the same units of measurements, also the gene names are presented with no apparent logic, and other unclear symbols are presented but not clarified.
  • In several places, the Authors present antibiotic and ARGs concentration in the environment as if they were one and the same thing.
  • Some presented papers are described in great detail with numbers, percentages, concentrations, etc., however, no conclusion or overall summary of research data is presented by the Authors, leaving the overall point of the information presented unclear.
  • Some information is not presented in adequate detail, or it is not decribed correctly (e.g. the overall structure and process of water treatment plants, the extracellular polymers and their role, their part in antibiotic resistance, integrons and DNA transfer tools used by bacteria, QS, antibiotic degradation vs. mineralisation etc.).
  • The scientific terms are not used correctly: gene names, bacterial species and genera should adhere to writing rules, terms like pili, quorum sensing, etc. are often incorrectly addressed.
  • The last sentences of the Conclusions present information that was not addressed in the manuscript at all.
Comments on the Quality of English Language

The English language is often confusing, and uses either very technical terms (when water treatment plant procedures are presented), or an overly simplified non-scientific view (e.g. calling pili "hairs" is not appropriate in an academic text).

Author Response

Reviewer #3

First, we would like to thank the reviewers and the editor for the positive and constructive comments and suggestions,we have carefully checked the manuscript and the modifications were noted in the revised manuscript with changes marked (in red).

  1. In the manuscript "The Environmental Lifecycle of Antibiotics and Resistance Genes: Transmission Mechanisms, Challenges, and Control Strategies", the Authors aim to present the fate of antibiotic resistance genes and bacteria after they leave the places where antibiotic treatment is performed, and transfer to the environment or treatment plants. The topic is interesting, and indeed deserves a thorough review, however, i have some major concerns with this manuscript:

Response:We appreciate the editor's accurate summary of the core objectives of the manuscript. Indeed, this study aims to elucidate the fate and transmission mechanisms of antibiotics, antibiotic-resistant bacteria (ARBs), and antibiotic resistance genes (ARGs) after their release from treatment sites into environmental media and wastewater treatment plants. I will respond to the questions raised below.

  1. The writing style is inconsistent. Some parts of the manuscript are too simple for a review article; e.g. in chapters 3.1-3.2, very basic information is presented, which does not fit well with the topic. It also does not cover all the necessary pathways that bacteria can exchange DNA. Other chapters, on the contrary, are heavy with technological terms, presenting water treatment plant operations and tools without clear explanations.

Response:Thanks. Sections 3.1-3.2 have been supplemented and analyzed in depth. Scenario analyses have been added for the four drug resistance mechanisms and the four HGT mechanisms, and new sections and content have been added. The following is the title of the third chapter after the additions:

3.1. The molecular basis of bacterial drug resistance

3.1.1. Intrinsic resistance

(1) Bacteria alter extracellular membrane permeability

(2) Active transport of bacteria via efflux pumps

(3) Bacterial modification of antibiotic target molecules

(4) Bacteria inactivate antibiotics through enzymes

3.1.2. Acquired drug resistance: gene mutations and horizontal transfer (HGT)

3.2 Environmental pressures driving and synergistically influencing the evolution of drug resistance

3.2.1 Environmental pressure as a driving force

(1) Antibiotics

(2) Heavy metals

(3) Disinfectant

3.2.2 Synergistic effects of environmental stresses

3.2.2.1 Co-selection resistance driven by multiple pollutants

(1) Co-resistance

(2) Cross-resistance

(3) Co-regulation

3.2.2.2 Regulatory network of quorum sensing (QS) system-regulated biofilm resistance

3.3. Critical pathways for horizontal gene transfer (HGT)

3.3.1 Mobile genetic elements

(1) Plasmids

(2) ICEs/IMEs

(3) Bacteriophages

(4) Transposons

(5) Integrons

3.3.2. Conjugation: the role of plasmids and integrons

3.3.3. Transformation: capture and integration of free DNA

3.3.4. Transduction: phage-mediated gene delivery

3.3.5. Gene delivery potential of outer membrane vesicles (OMVs)

  1. Not all information presented in the text has appropriate references; some references do not contain information that the Authors claim.

Response:Thanks, after receiving this comment, we completely re-examined the citation of the paper and changed the citation below the original

Line38-43,modified reference 2

"Between 2000 and 2015, global antibiotic consumption increased from 2.11 billion tonnes to 3.48 billion tonnes, and it is projected to increase by 67% by 2030, with emerging economies such as China and India contributing to the major incremental increase [2, 3]."Changed to "From 2016 to 2023, global antibiotic consumption in major countries increased from 2.95 billion defined daily doses (DDD) to 34.3 billion DDD, representing a 16.3% increase. This reflects a rise in consumption rates from 13.7 DDD per 1,000 residents per day to 15.2 DDD per 1,000 residents per day, an increase of 10.6%. By 2030, it is projected to rise to 751 billion DDD, representing a 52.3% increase [2]."

  1. The information in figures and tables is quite confusing - figure 1 does not have yellow or green lines referenced in the text, in figures 2-3 the arrows are not of the correct orientation, the tables present a lot of data, however, it is not of the same units of measurements, also the gene names are presented with no apparent logic, and other unclear symbols are presented but not clarified.

Response:Thanks. We have changed the yellow-green lines in the text to red-blue lines. Figure 2-3 has been re-uploaded. The gene names in Table 1 have been standardized, and the correct format has been applied to all gene names. Due to the lack of a unified standard across different literature for different units of genes, this also posed challenges for our work. Absolute quantification uses copies/mL and copies/g, while relative quantification studies use ARG/16S rRNA and log(copies/ng DNA), among other formats. We have provided additional explanations in the original text regarding this issue.

"Figure 1 depicts the cycle of ARGs in the environment. The blue lines represent the transmission path of ARGs from humans to the environment, while the red lines represent the cycle of ARGs back to humans."

"bacitracin and sulfonamide resistance genes"changed to TEM,sul1,sul2

Additional information"Absolute quantitative studies use units such as copies/mL and copies/g.Relative quantitative studies use forms such as ARG/16S rRNA and log(copies/ng DNA)."

  1. In several places, the Authors present antibiotic and ARGs concentration in the environment as if they were one and the same thing.

Response:Thanks, There is indeed a cross-reference between antibiotic concentration and ARG concentration in the article. We have confirmed and changed both based on the units.

  1. Some presented papers are described in great detail with numbers, percentages, concentrations, etc., however, no conclusion or overall summary of research data is presented by the Authors, leaving the overall point of the information presented unclear.

Response:Thanks. We have provided additional analysis for both tables in the original text to address this issue.

"From the table, we can also see that hospital wastewater and pharmaceutical wastewater are “super hotspots” for ARGs, especially high-risk genes such as blaOXA-48 and qacE."

"It can be seen that MBR is the most effective in water treatment, but membrane fouling needs to be controlled. Sulfonamides are currently the most difficult type of resistant genes to treat. The treatment effects of different treatment technologies are not satisfactory, and most of the removal still causes secondary pollution."

  1. Some information is not presented in adequate detail, or it is not decribed correctly (e.g. the overall structure and process of water treatment plants, the extracellular polymers and their role, their part in antibiotic resistance, integrons and DNA transfer tools used by bacteria, QS, antibiotic degradation vs. mineralisation etc.).

Response:Thanks, We have added "Mobile genetic elements" in Chapter 3, line 647-694,which provides a detailed introduction to the DNA transfer tools used by bacteria.

  1. The scientific terms are not used correctly: gene names, bacterial species and genera should adhere to writing rules, terms like pili, quorum sensing, etc. are often incorrectly addressed.

Response:Thanks, the scientific terms have been corrected. Bacteria and genes are now italicized with the first letter capitalized.

  1. The last sentences of the Conclusions present information that was not addressed in the manuscript at all.

Response:Thanks We have revised and supplemented the conclusion and outlook section. We have deleted the original content from 876 to 900 and added three new points for discussion:

(1)Gaps in toxicity assessment of intermediate metabolites line 1151-1163

(2)Limitations of detection technologies and gaps in dynamic monitoring line 1164-1180

(3)Limitations of detection technologies and gaps in dynamic monitoring line1181-1197

  1. 10. The English language is often confusing, and uses either very technical terms (when water treatment plant procedures are presented), or an overly simplified non-scientific view (e.g. calling pili "hairs" is not appropriate in an academic text).

Response:Thanks,we have changed "pili" in the original text to "fimbriae."

Reviewer 4 Report

Comments and Suggestions for Authors

Dear Authors,

Thank you for the opportunity to review your manuscript, which explores a highly relevant and timely topic—the environmental lifecycle of antibiotic resistance genes (ARGs). The manuscript is ambitious in scope, addressing a broad range of sources, mechanisms, and control strategies associated with the dissemination and mitigation of ARGs.

To support the further development of your work, I have outlined below a series of constructive comments and suggestions aimed at enhancing the manuscript’s clarity, scientific rigor, and overall scholarly impact.

1. Scientific Accuracy and Credibility

The statement reporting an increase in antibiotic use from “2.11 billion tonnes to 3.48 billion tonnes” appears implausible and likely stems from a unit conversion error. Such inaccuracies can significantly compromise the manuscript’s credibility and should be corrected or reformulated using verifiable data and appropriate citations.

2. Structure and Organization

Although the manuscript follows a generally logical structure, transitions between major sections—particularly from environmental processes to biomedical applications—are often abrupt. Strengthening the narrative cohesion with transitional framing would greatly improve the reader’s experience.
In addition, internal redundancies, such as repeated descriptions of resistance mechanisms (e.g., efflux pumps, enzymatic inactivation), should be streamlined to enhance conciseness and avoid repetition.

3. Depth and Critical Analysis

Several sections, notably those discussing ARG migration and treatment technologies, remain largely descriptive. Greater analytical depth is needed. For example:

  • The environmental fate and persistence of ARGs are mentioned but not examined through kinetic modeling or concentration-dependent frameworks.

  • The discussion of removal strategies lacks evaluation of associated trade-offs, such as potential for ARG reactivation or the economic and operational feasibility of advanced treatment systems.
    Incorporating comparative performance metrics and real-world applications—such as case studies or pilot-scale data—would significantly strengthen the manuscript’s relevance and applicability.

4. Language and Expression

The manuscript would benefit from substantial language revision. Numerous sentences are overly long, awkwardly phrased, or grammatically inconsistent, which can obscure the intended meaning.
It is strongly recommended that the text be professionally edited by a native English-speaking scientific editor to ensure clarity, fluency, and adherence to academic conventions.                  

5. Use of References and Citations

While the manuscript is well-cited, the use of references often lacks analytical integration. Citations are frequently grouped without adequate synthesis or critical commentary. Enhancing the interpretive use of the literature—by highlighting areas of consensus, divergence, or innovation—would improve the scholarly contribution of the review.

6. Figures and Data Presentation

Several figures (e.g., Figure 2) are referenced but not sufficiently discussed or contextualized within the narrative. To maximize their value, each figure should be clearly integrated and its relevance explicitly explained.

7. Conclusions and Future Directions

The conclusion would benefit from a more strategic, forward-looking structure. At present, it largely reiterates earlier points without offering synthesis or prioritization.
Moreover, the review would be strengthened by a candid reflection on its own limitations—such as geographic constraints, lack of quantitative analysis, or omission of emerging ARG vectors (e.g., viruses).

Final Remarks

Your manuscript addresses a complex and increasingly urgent issue at the intersection of environmental microbiology and public health. With thoughtful revision—particularly in terms of scientific precision, thematic cohesion, and language refinement—it holds the potential to make a meaningful contribution to the field of antimicrobial resistance.

I trust these observations will assist you in strengthening your work, and I wish you continued success in your important research endeavors.

Warm regards,
Reviewer

Comments on the Quality of English Language

The English could be improved to more clearly express the research.

Author Response

Reviewer #4

  1. Scientific Accuracy and Credibility

The statement reporting an increase in antibiotic use from “2.11 billion tonnes to 3.48 billion tonnes” appears implausible and likely stems from a unit conversion error. Such inaccuracies can significantly compromise the manuscript’s credibility and should be corrected or reformulated using verifiable data and appropriate citations.

Response:Thanks, we have rechecked the content and made changes to the data and replaced the references.Changed to "From 2016 to 2023, global antibiotic consumption in major countries increased from 2.95 billion defined daily doses (DDD) to 34.3 billion DDD, representing a 16.3% increase. This reflects a rise in consumption rates from 13.7 DDD per 1,000 residents per day to 15.2 DDD per 1,000 residents per day, an increase of 10.6%. By 2030, it is projected to rise to 751 billion DDD, representing a 52.3% increase [2]."

  1. Structure and Organization

Although the manuscript follows a generally logical structure, transitions between major sections—particularly from environmental processes to biomedical applications—are often abrupt. Strengthening the narrative cohesion with transitional framing would greatly improve the reader’s experience.
In addition, internal redundancies, such as repeated descriptions of resistance mechanisms (e.g., efflux pumps, enzymatic inactivation), should be streamlined to enhance conciseness and avoid repetition.

Response:Thanks. We have added a new subsection titled "Detection and Quantitative Analysis of Resistance Genes in the Environment" in Chapter 2 to serve as a transition. We have also replaced the repetitive simple descriptions in Chapter 3 with a more in-depth analysis of environmental scenarios, added a new subsection titled "Synergistic Effects Under Environmental Stress," and supplemented the content with "Mobile Genetic Elements."

  1. Depth and Critical Analysis

Several sections, notably those discussing ARG migration and treatment technologies, remain largely descriptive. Greater analytical depth is needed. For example:

The environmental fate and persistence of ARGs are mentioned but not examined through kinetic modeling or concentration-dependent frameworks.

The discussion of removal strategies lacks evaluation of associated trade-offs, such as potential for ARG reactivation or the economic and operational feasibility of advanced treatment systems.
Incorporating comparative performance metrics and real-world applications—such as case studies or pilot-scale data—would significantly strengthen the manuscript’s relevance and applicability.

Response:Thanks. In response to this issue, we have replaced the simple description repeated in Chapter 3 with a more in-depth analysis of the environmental context, added a new subsection on "synergistic effects under environmental stress," line 536-645 and supplemented the content with "mobile genetic elements."line 647-694

  1. Language and Expression

The manuscript would benefit from substantial language revision. Numerous sentences are overly long, awkwardly phrased, or grammatically inconsistent, which can obscure the intended meaning.
It is strongly recommended that the text be professionally edited by a native English-speaking scientific editor to ensure clarity, fluency, and adherence to academic conventions.

Response:Thanks, for bringing this issue to our attention. We have replaced several overly long and cumbersome sentences in the article.

  1. Use of References and Citations

While the manuscript is well-cited, the use of references often lacks analytical integration. Citations are frequently grouped without adequate synthesis or critical commentary. Enhancing the interpretive use of the literature—by highlighting areas of consensus, divergence, or innovation—would improve the scholarly contribution of the review.

Response:Thanks, Regarding this issue, although we did not conduct further citation management, we replaced some references that were not relevant to the original text and were outdated during the review process.

Line42-43Modified reference 2

Line267-268Modified reference 67

  1. Figures and Data Presentation

Several figures (e.g., Figure 2) are referenced but not sufficiently discussed or contextualized within the narrative. To maximize their value, each figure should be clearly integrated and its relevance explicitly explained.

Response:Thanks. We have provided additional analysis for both tables in the original text to address this issue.

"From the table, we can also see that hospital wastewater and pharmaceutical wastewater are “super hotspots” for ARGs, especially high-risk genes such as blaOXA-48 and qacE."

"It can be seen that MBR is the most effective in water treatment, but membrane fouling needs to be controlled. Sulfonamides are currently the most difficult type of resistant genes to treat. The treatment effects of different treatment technologies are not satisfactory, and most of the removal still causes secondary pollution."

  1. Conclusions and Future Directions

The conclusion would benefit from a more strategic, forward-looking structure. At present, it largely reiterates earlier points without offering synthesis or prioritization.
Moreover, the review would be strengthened by a candid reflection on its own limitations—such as geographic constraints, lack of quantitative analysis, or omission of emerging ARG vectors (e.g., viruses).

Response:Thanks We have revised and supplemented the conclusion and outlook section. We have deleted the original content from 876 to 900 and added three new points for discussion:

(1)Gaps in toxicity assessment of intermediate metabolites line 1151-1163

(2)Limitations of detection technologies and gaps in dynamic monitoring line 1164-1180

(3)Limitations of detection technologies and gaps in dynamic monitoring line 1181-1197

Final Remarks

Your manuscript addresses a complex and increasingly urgent issue at the intersection of environmental microbiology and public health. With thoughtful
revision—particularly in terms of scientific precision, thematic cohesion, and language refinement—it holds the potential to make a meaningful contribution to the field of antimicrobial resistance. I trust these observations will assist you in strengthening your work, and I wish you continued success in your important research endeavors.

Response:We sincerely appreciate the Editor's thoughtful assessment of our
manuscript and the encouraging comments regarding its potential contribution to antimicrobial resistance research. We are grateful for the constructive feedback on enhancing scientific precision, thematic cohesion, and language refinement. We will
carefully address all points raised during the revision process to strengthen the work. Thank you for supporting this important endeavor.

Reviewer 5 Report

Comments and Suggestions for Authors

A detailed assessment with specific comments is provided in the attached file.

Author Response

Reviewer #5

First, we would like to thank the reviewers and the editor for the positive and constructive comments and suggestions,we have carefully checked the manuscript and the modifications were noted in the revised manuscript with changes marked (in red).

  1. Incorrect use of italics for scientific names and gene symbols

Throughout the manuscript, there is inconsistent formatting of scientific names and gene symbols:

  • Bacterial species names (e.g., Escherichia coli, Klebsiella pneumoniae, Pseudomonas aeruginosa) are not italicised as required by international taxonomic standards.
  • Gene names (e.g., tetA, sul1, blaCTX-M) are frequently presented in plain text, instead of italics. Please revise the entire manuscript carefully to conform to standard formatting conventions for microbial species and genetic nomenclature.

Response:Thanks.We have revised the font used for the words “microorganism” and “gene” in the article one by one to address this issue.

  1. Overly generic description of antimicrobial resistance mechanisms

Section 3.1 presents the canonical mechanisms of antimicrobial resistance (e.g., target modification, efflux pumps, enzymatic inactivation, membrane permeability) in a textbook-like manner. While factually correct, this content is:

  • Widely documented in existing literature; • Descriptive rather than analytical; • Lacking in novel insight or environmental contextualisation.

To better serve the scope of this review, I recommend narrowing the section and shifting the focus toward:

  • The prevalence and functional expression of these resistance mechanisms in environmental strains;
  • Experimental or metagenomic studies that demonstrate these mechanisms in situ (e.g., in WWTPs, agricultural soils, aerosols);
  • Differences in resistance mechanism dominance across clinical vs. environmental settings.

Response:Thanks. In the discussion in 3.1, we have replaced the original simple and broad content with an analysis of environmental scenarios under different drug resistance mechanisms.

Line 405-413,420-428,439-445,462-469,

  1. Horizontal gene transfer (HGT) mechanisms: lack of environmental anchoring

The explanation of HGT (conjugation, transformation, transduction, OMVs) is comprehensive but remains abstract and overly general. The manuscript would benefit significantly from grounding this discussion in environmental contexts:

  • Please cite studies that have demonstrated these HGT mechanisms in natural environments, such as wastewater treatment plants, contaminated soils, river sediments, and farm runoff.
  • Where possible, specify which ARGs and mobile genetic elements were involved, and under what ecological pressures they were mobilised.
  • Consider summarising HGT events by environment type to reinforce the environmental dimension of the review.

Currently, the section lacks evidence-driven examples and risks overlapping with other existing reviews without offering added value.

Response:Thanks. We have added a subsection on heritable elements in Chapter 3. In the subsequent discussion of HGT, we have removed the original simple and general content and added an analysis of the environmental scenarios under each mechanism.

Line 715-731,750-762,783-800,818-832

  1. Weak integration between environmental and molecular sections

There is a noticeable disconnect between the early sections describing the environmental lifecycle of antibiotics and ARGs and the later molecular biology-focused sections. The manuscript would benefit from:

  • Explicitly linking molecular resistance and transfer mechanisms to previously discussed environmental compartments;
  • Highlighting, for example, how biofilm formation and ARG transfer via integrons is potentiated in WWTPs;
  • Mapping environmental pressures (e.g., heavy metals, nutrient loads) to specific resistance pathways.

Integrating these levels of analysis will provide a more coherent and interdisciplinary narrative.

Response:Thanks. In response to this issue, we have added a new section titled "Detection and Quantitative Analysis of Resistance Genes in the Environment"line 328-378 at the end of Chapter 2. This section introduces the detection of resistance genes in the environment and leads into the subsequent discussion on the molecular mechanisms of resistance genes.

5.Repetition and redundancy of core concepts

Several concepts are repeated verbatim or near-verbatim across the manuscript (e.g., the role of WWTPs as ARG “amplifiers”, EPS barrier functions, HGT processes), which inflates the text unnecessarily and disrupts clarity.

I recommend a critical synthesis of previously mentioned points instead of reiteration, and consolidation of overlapping sections to improve readability and coherence.

Response:Thanks, We have replaced repeated concepts with abbreviations in the article. We have also deleted some concepts, such as “sewage treatment plants as ARGs amplifiers.”

  1. Limited critical evaluation of the literature

The manuscript often reports findings from other studies without analytical commentary. For example:

  • There is little discussion about the methodological limitations or comparative strengths of cited studies;
  • Conflicting findings, if any, are not highlighted;
  • No reflection is provided on the reliability or transferability of laboratory-based studies to environmental conditions.

Adding a critical layer to the literature appraisal would strengthen the scientific value of the review and demonstrate a more comprehensive understanding of the field.

Response:Thanks. In response to this issue, we have added comparisons between clinical and natural environments to the new supplementary content on the mechanisms of bacterial resistance. In the conclusion and outlook section, we have also included an analysis and critique of existing research.

  1. Insufficient use of recent or high-impact literature

Many references cited are dated (>5 years old) or drawn from general background sources. The manuscript would benefit from inclusion of:

  • Recent metagenomic and high-throughput studies that reveal ARG dynamics in real environmental samples;
  • Large-scale environmental surveillance data;
  • Systematic reviews or meta-analyses relevant to ARG spread and mitigation. This would increase the impact and relevance of the review, particularly in the context of environmental and public health policy.

Response:Thanks. In response to this issue, we have emphasized the use of high-impact factor literature from the past five years in a series of newly added content. The newly added chapters include:

2.1.4Major antibiotic families and ARG subtypes in the environment

2.4Detection and quantification of resistance genes in the environment

3.2 Environmental pressures driving and synergistically influencing the evolution of drug resistance

3.3.1 Mobile genetic elements

  1. Missing discussion of knowledge gaps

The review does not sufficiently identify or discuss open questions or gaps in current knowledge.

Please consider including a dedicated subsection that addresses:

  • Unresolved mechanisms in environmental ARG transmission;
  • Limitations in current ARG surveillance and quantification techniques;
  • Challenges in connecting environmental ARGs to clinical resistance profiles Such a section is essential in a comprehensive review and provides valuable guidance for future research.

Response:Thanks We have revised and supplemented the conclusion and outlook section. We have deleted the original content from 876 to 900 and added three new points for discussion:

(1)Gaps in toxicity assessment of intermediate metabolites line 1151-1163

(2)Limitations of detection technologies and gaps in dynamic monitoring line 1164-1180

(3)Limitations of detection technologies and gaps in dynamic monitoring line 1181-1197

  1. Underdeveloped treatment of emerging solutions (e.g., crispr, nanomaterials) Although emerging technologies such as CRISPR-Cas systems and nanomaterials are briefly mentioned in the introduction, they are not developed later in the manuscript. Given their potential in ARG mitigation, a more substantial discussion of:
  • CRISPR-based gene editing for ARG removal;
  • Antimicrobial nanomaterials for water treatment;
  • Phage therapy or microbiome engineering approaches;

would enhance the novelty and practical relevance of the review.

Response:Thanks. In response to this issue, we have added a subsection titled "Emerging Technologies"line 1010-1088  in Chapter 4 to provide a detailed introduction to the applications of CRISPR-Cas gene editing technology and nanomaterial technology.

  1. Weak visual communication and data synthesis

The manuscript relies heavily on dense text and long tables but lacks:

  • Visual summaries or schematics showing transmission pathways or molecular interactions;
  • Comparative diagrams of ARG removal efficiency by treatment technology;
  • Conceptual models connecting anthropogenic drivers to molecular outcomes.

I strongly encourage the inclusion of 2–3 concise, well-labelled figures or summary charts to aid reader comprehension.

Response:We sincerely appreciate the reviewers' constructive suggestions for improving visual communication. While we recognise the value of diagrams in summarising complex processes (such as ARG transmission pathways and treatment technology efficiency), the current revision schedule does not allow for the creation of new diagrams without compromising scientific rigour. We have merged and modified the data in the existing tables.

  1. Terminological issues and inconsistencies

Several important inconsistencies in terminology and scientific accuracy were noted:

  • Terms such as “ARB” (antibiotic-resistant bacteria) and “MDRB” (multidrug-resistant bacteria) are used interchangeably, despite denoting different phenomena. Please clarify the intended usage and distinguish where appropriate. • The term “sulforaphane ARGs” (Section 2.2.2) appears to be a significant error. Sulforaphane is a dietary phytochemical with no known relation to antibiotic resistance genes. The accompanying claim about its concentration in edible crops is also unsupported by the cited reference ([45]), which does not mention sulforaphane, sulfonamides, or ARGs in plant tissues. This should be corrected to refer — if appropriate — to sulfonamide ARGs, with a verified and relevant source.

Please revise the text to ensure terminological precision and biological plausibility

Response:Thanks. We have replaced MDRB with "ARB" throughout the document, corrected the terminology error in section 2.2.2, and replaced the references."sulforaphane ARGs"changed to "sulfonamide ARGs"

  1. Reference list does not comply with MDPI formatting guidelines

The references currently do not follow MDPI’s formatting requirements. Specifically:

  • No DOI is provided for any of the references, despite DOIs being mandatory where available;
  • Scientific species names and genes are not italicised in the reference list;
  • Journal titles are not formatted in italics or abbreviated according to MDPI guidelines.

Please revise the entire reference section according to the journal’s reference style guide. Accurate, complete and properly formatted citations are essential for scientific reproducibility and formal publication.

Response:Thanks. We have revised the reference format in response to this issue.

In summary, the manuscript addresses an important topic with broad implications for environmental and public health. However, it requires substantial revision in structure, critical depth, environmental contextualisation, and formatting to meet the standards of a high-impact review article.

Response:We sincerely appreciate the Editor's thoughtful assessment of our manuscript and the encouraging comments regarding its potential contribution to antimicrobial resistance research. We are grateful for the constructive feedback on enhancing scientific precision, thematic cohesion, and language refinement. We will carefully address all points raised during the revision process to strengthen the work. Thank you for supporting this important endeavor.

Round 2

Reviewer 1 Report

Comments and Suggestions for Authors

After considering the revised manuscript document by the authors, it is very clear that a much more mature version has been created and it is ready for publication.

The authors have taken into account the suggestions made and corrected the minor details pointed out to enhance the quality of the manuscript. This will ensure that readers find solid and useful material for their research and academic activities in the field.

It only point that remains to note is that in "Table 1", Section "Livestock wastewater", tetC, and tetO have to be also in italics.Congratulations to the authors for this excellent work and many thanks to the editors for considering my expertise in the field to evaluate the work of other colleagues.

Best regards

Author Response

Response to the reviewers’ comments

First, we would like to thank the reviewers and the editor for the positive and constructive comments and suggestions, we have carefully checked the manuscript and the modifications were noted in the revised manuscript with changes marked (in red).

(1)After considering the revised manuscript document by the authors, it is very clear that a much more mature version has been created and it is ready for publication.

The authors have taken into account the suggestions made and corrected the minor details pointed out to enhance the quality of the manuscript. This will ensure that readers find solid and useful material for their research and academic activities in the field.

It only point that remains to note is that in "Table 1", Section "Livestock wastewater", tetC, and tetO have to be also in italics.Congratulations to the authors for this excellent work and many thanks to the editors for considering my expertise in the field to evaluate the work of other colleagues.

Response:Thank you. We have uniformly changed the gene names in the “Livestock and Poultry Wastewater” section of the table to italics. Additionally, we have replaced ''content'' in the table header with the more accurate term ''Relative abundance''

Source

ARGs

Relative abundance (copies/16S rRNA gene copies)

References

Sludge sampled from municipal wastewater treatment plant

tetA, tetB, tetE, tetG, tetH, tetS, tetT, tetX, sul1, sul2, qnrB, and ermC

(1.5±2.3)×109 - (2.2±2.8)×1011

copies/ g dry weight

[20]

Municipal wastewater

tetA, tetC, tetG, tetM, tetO, tetW, tetX, sul1, sul2

3.6×101(teW) to 5.4×106

(tetX) copies mL-1

6.4×1012(tetW)to 1.7×1018

(sull) copics d-1

[21]

Sludge sampled from hospital wastewater treatment plant

blaOXA-48CTX-MblaIMP blaTEM

5.36×1011 - 1.90×1012

copies/ g dry weight

[22]

Hospital wastewater

sul1, blaSHV, catA1 ,aacC2, tetA;

1.94×101,4.39×10-3,6.83×10-5,5.67×10-3,3.46×10-3

[23]

sul1, sul2, sul3, tetQ;

1.79×101~6.67×101,7.33×10-2~3.38×101,9.22×10-2~5.9×101,2.8×101~7.47×101;

[24]

Livestock wastewater

tetL, strB, sul2, tetG, ermB, sul1, tetX,and cmlA

tetL(1.36~0.39), strB (0.82~0.52), sul2 (0.96~0.64), tetG (1.81~0.67), ermB (1.17~0.71), sul1 (1.51~0.93), tetX (1.17~0.94), and cmlA(1.73~1.14)

[25]

tetX, ermF, ermB, mefA, tetM, sul2

2.43×1011- 5.69×1010

copies/mL

[26]

sul1, sul2, tetM;

3.84×101, 1.62×101, 2.33×101

[27]

tetC, tetO

7.3×103, 1.7×101

[21]

Pharmaceutical industries wastewater

tetA, tetC, tetG, tetL, tetM, tetO,

1.4×101, 3.2×102, 5.1×102, 6.1×102, 1.1×102, 1.0×100, 1.8×100, 1.6×101, 3.7×103

[28]

sul1, sul2, tetA, qacE, qacED1;

101 to 102

[29]

Soil irrigated with recycled water

tetG, tetW, sulI, sulII, intI1

Highest abundance of sul2 and intI1, the abundances−71 were 8.43×107 copies g-1 dry soil, 7.62×107 copies g-1 dry soil

[30]

Drinking water

sul1, sul2, tetC, tetG, tetX, tetA, tetB, tetO, tetM, and tetW

Total concentrations of ARGs belonging to either the sulfonamide or tetracycline resistance gene class were above 105 copies mL-1

[31]

River sediments

TEM,sul1,sul2

1.09 × 10−1

~1.06 × 10−1

[32]

Reviewer 3 Report

Comments and Suggestions for Authors

To my understanding, the aim of the manuscript is to present the fate and transformation of antibiotics and antibiotic resistance genes in the environment, as is now quite clearly addressed in the Abstract and the Introduction (after the revision). Sadly, I cannot say that the manuscript itself was consistently improved after the Author's revision. The writing style and depth of analysis remain inconsistent; not all the statements have appropriate references, there are incorrect conclusions throughout the text, only part of the incorrect scientific terms were corrected, and other parts remain. 

Comments on the Quality of English Language

English is mostly fine, but scientific terms are still used incorrectly. 

Author Response

Response to the reviewers’ comments

First, we would like to thank the reviewers and the editor for the positive and constructive comments and suggestions,we have carefully checked the manuscript and the modifications were noted in the revised manuscript with changes marked (in red).

To my understanding, the aim of the manuscript is to present the fate and transformation of antibiotics and antibiotic resistance genes in the environment, as is now quite clearly addressed in the Abstract and the Introduction (after the revision). Sadly, I cannot say that the manuscript itself was consistently improved after the Author's revision. The writing style and depth of analysis remain inconsistent; not all the statements have appropriate references, there are incorrect conclusions throughout the text, only part of the incorrect scientific terms were corrected, and other parts remain. 

Response:Thanks, We have made the following modifications in response to your concerns.

We have replaced the term ''metabolites'' with ''degradation products'' in lines 44, 77, 973, 1221, 1225, 1228, 1231, 1234, 1241, 1268, and 1308 of the text.

We have changed ''content'' in Table 1 to the more accurate term ''Relative abundance.''

We have amended the phrasing in line 503 from ''ARGs are usually encoded in MGEs'' to the more accurate statement ''ARGs are usually carried by MGEs''.

We have amended the term ''splice transfer'' in line 659 to the more precise expression ''conjugative transfer''.

We have revised the statement in line 1138 from ''thereby blocking the key pathyways of HGT at the source'' to the more accurate phrasing: ''thereby potentially reducing the frequency of HGT by degrading the genetic material available for transfer.''

Reviewer 4 Report

Comments and Suggestions for Authors

General Comments

The manuscript offers a thorough and valuable review of the sources, environmental transmission routes, molecular mechanisms, and control strategies associated with antibiotic resistance genes (ARGs). The integrative framework presented is both relevant and important, particularly in light of the growing global concern over antimicrobial resistance.

The manuscript is well-organized, with a logical progression from environmental origins and dissemination to resistance mechanisms and treatment approaches. The figures and tables effectively support the content and are well incorporated throughout.

The work is scientifically robust and well-supported by current and pertinent literature. The explanations of intrinsic and acquired resistance mechanisms are clear, and the section on horizontal gene transfer is especially comprehensive.

Suggestions

1) Although the manuscript is generally clear, many sentences are overly long or complex. Streamlining the language and refining sentence structure would significantly improve readability.
It is recommended to consider professional English editing to enhance fluency, grammar, and consistency—particularly in the abstract and introduction.

2) Some technical terms, such as “outer membrane vesicles” and “quorum sensing,” would benefit from clearer definitions to aid understanding among interdisciplinary readers.
Several sections could also be made more concise, especially where similar ideas are repeated.

3) Figure 1 effectively conveys ARG transmission pathways; however, the legend should be expanded to clarify the meaning of symbols and color codes.
Table 1 contains valuable information, but its clarity could be improved by standardizing units and enhancing formatting.

4) The section on CRISPR-Cas and nanomaterials is relatively brief. Expanding on their practical applications or incorporating recent examples would add depth and relevance.

5)The conclusion would benefit from a clearer summary of the key findings and a stronger emphasis on critical research and policy needs.
Adding a final paragraph outlining recommendations for future research or potential interdisciplinary collaboration would further enhance the manuscript’s impact.

Author Response

Response to the reviewers’ comments

First, we would like to thank the reviewers and the editor for the positive and constructive comments and suggestions,we have carefully checked the manuscript and the modifications were noted in the revised manuscript with changes marked (in red).

General Comments

The manuscript offers a thorough and valuable review of the sources, environmental transmission routes, molecular mechanisms, and control strategies associated with antibiotic resistance genes (ARGs). The integrative framework presented is both relevant and important, particularly in light of the growing global concern over antimicrobial resistance.

The manuscript is well-organized, with a logical progression from environmental origins and dissemination to resistance mechanisms and treatment approaches. The figures and tables effectively support the content and are well incorporated throughout.

The work is scientifically robust and well-supported by current and pertinent literature. The explanations of intrinsic and acquired resistance mechanisms are clear, and the section on horizontal gene transfer is especially comprehensive.

Response:Thank you for your positive and encouraging feedback on our manuscript. We greatly appreciate your recognition of the review’s comprehensive scope, logical organization, and robust scientific analysis (especially regarding resistance mechanisms and HGT). Your comments motivate us to continue contributing to this critical field.

Suggestions

(1) Although the manuscript is generally clear, many sentences are overly long or complex. Streamlining the language and refining sentence structure would significantly improve readability.
It is recommended to consider professional English editing to enhance fluency, grammar, and consistency—particularly in the abstract and introduction.

Response: Thanks, we have revised the abstract and introduction sections (line35-70) and simplified their wording.

Abstract:

''Antibiotics are widely used in modern medicine. However, as global antibiotic consumption rises, environmental contamination with antibiotics and antibiotic resistance genes (ARGs) is becoming a serious concern. The impact of antibiotic use on human health is now under scrutiny, particularly regarding the emergence of antibiotic-resistant bacteria (ARB) in the environment. This has heightened interest in technologies for treating ARGs, highlighting the need for effective solutions. This review traces the life cycle of ARBs and ARGs driven by human activity, revealing pathways from antibiotic use to human infection. We address the mechanisms enabling resistance in ARBs during this process. Beyond intrinsic resistance, the primary cause of ARB resistance is the horizontal gene transfer (HGT) of ARGs. These genes exploit mobile genetic elements (MGEs) to spread via conjugation, transformation, transduction, and outer membrane vesicles (OMVs). Currently, biological wastewater treatment is the primary pollution control method due to its cost-effectiveness. However, these biological processes can promote ARG propagation, significantly amplifying the environmental threat posed by antibiotics. This review also summarizes key mechanisms in the biological treatment of antibiotics and evaluates risks associated with major ARB/ARG removal processes. Our aim is to enhance understanding of ARB risks, their pathways and mechanisms in biotreatment, and potential biomedical applications for pollution control.''

In the part of Introduction:

''The discovery and use of antibiotics marked a turning point in medicine, significantly reducing mortality from bacterial infections [1]. However, global population growth and rising healthcare needs have exacerbated antibiotic misuse. Between 2016 and 2023, global antibiotic consumption in major countries surged from 2.95 billion to 34.3 billion defined daily doses (DDD), a 16.3% increase. Consumption rates rose from 13.7 to 15.2 DDD per 1,000 residents per day (a 10.6% increase) and are projected to reach 751 billion DDD by 2030, representing a 52.3% rise [2].………………This paper aims to develop a comprehensive 'environmental cycle-molecular mechanism-technological control' analytical framework. Our objectives are to reveal the cross-media transmission pathways of ARGs, evaluate the efficiency of biological treatment processes, and propose multidisciplinary management strategies to help mitigate the global antibiotic resistance crisis.''

(2)Some technical terms, such as “outer membrane vesicles” and “quorum sensing,” would benefit from clearer definitions to aid understanding among interdisciplinary readers.Several sections could also be made more concise, especially where similar ideas are repeated.

Response: Thanks, we have added definitions and explanations for these terms in the original text as follows: ''Quorum sensing is a cellular communication mechanism by which microorganisms monitor population density and trigger collective behavior through the secretion and perception of specific signaling molecules. As the bacterial population grows and the concentration of signaling molecules in the environment reaches a threshold, they bind to intracellular receptor proteins to form complexes (such as LuxR-type transcription factors). This process activates the synchronized expression of target genes and coordinates the regulation of collective phenotypes. Examples of these phenotypes include biofilm formation, virulence factor secretion, and horizontal transfer of antibiotic resistance genes.''(line59-607)

''Outer membrane vesicles (OMVs) are nanoscale (20–250 nm) bilayer lipid structures released by Gram-negative bacteria through outer membrane budding. The composition of their membranes is identical to that of the parent bacterial outer membrane (containing lipopolysaccharide LPS and outer membrane proteins). These membranes encapsulate proteins, RNA, and DNA fragments (including chromosomal DNA, plasmids, and antibiotic resistance genes, or ARGs) from the periplasmic space. In the context of horizontal gene transfer (HGT), OMVs function as non-contact gene delivery vehicles, facilitating the transfer of genetic material across species boundaries to recipient bacteria (same or different species) or eukaryotic cells via membrane fusion or endocytosis .''(line814-822)

(3)Figure 1 effectively conveys ARG transmission pathways; however, the legend should be expanded to clarify the meaning of symbols and color codes.
Table 1 contains valuable information, but its clarity could be improved by standardizing units and enhancing formatting.

Response: Thanks, we have added a legend in Figure 1.

Figure 1. Life cycle of antibiotic resistance genes in the environment.

(4) Table 1 contains valuable information, but its clarity could be improved by standardizing units and enhancing formatting.

Response: Thanks, we have improved Table 1 and Table 2

Table 1. Abundance of ARGs in different sources of WWTP.

Source

ARGs

Content (copies/16S rRNA gene copies)

References

Sludge sampled from municipal wastewater treatment plants sludge

tetA, tetB, tetE, tetG, tetH, tetS, tetT, tetX, sul1, sul2, qnrB, and ermC

(1.5±2.3)×109 - (2.2±2.8)×1011

copies/ g dry weight

[20]

Municipal wastewater

tetA, tetC, tetG, tetM, tetO, tetW, tetX, sul1, sul2

3.6×101(teW) to 5.4×106

(tetX) copies mL-1

6.4×1012(tetW)to 1.7×1018

(sull) copics d-1

[21]

Sludge sampled from hospital wastewater treatment plants sludge

blaOXA-48,CTX-M,blaIMP blaTEM

5.36×1011 - 1.90×1012

copies/ g dry weight

[22]

Hospital wastewater

sul1, blaSHV, catA1 ,aacC2, tetA;

1.94×101,4.39×10-3,6.83×10-5,5.67×10-3,3.46×10-3

[23]

sul1, sul2, sul3, tetQ;

1.79×101~6.67×101,7.33×10-2~3.38×101,9.22×10-2~5.9×101,2.8×101~7.47×101;

[24]

Livestock wastewater

tetL, strB, sul2, tetG, ermB, sul1, tetX,and cmlA

tetL(1.36~0.39), strB (0.82~0.52), sul2 (0.96~0.64), tetG (1.81~0.67), ermB (1.17~0.71), sul1 (1.51~0.93), tetX (1.17~0.94), and cmlA(1.73~1.14)

[25]

tetX, ermF, ermB, mefA, tetM, sul2

2.43×1011- 5.69×1010

copies/mL

[26]

sul1, sul2, tetM;

3.84×101, 1.62×101, 2.33×101

[27]

tetC, tetO

7.3×103, 1.7×101

[21]

Pharmaceutical industries wastewater

tetA, tetC, tetG, tetL, tetM, tetO,

1.4×101, 3.2×102, 5.1×102, 6.1×102, 1.1×102, 1.0×100, 1.8×100, 1.6×101, 3.7×103

[28]

sul1, sul2, tetA, qacE, qacED1;

101 to 102

[29]

Soil irrigated with recycled water 

tetG, tetW, sulI, sulII, intI1

Highest abundance of sul2 and intI1, the abundances−71 were 8.43×107 copies g-1 dry soil, 7.62×107 copies g-1 dry soil

[30]

Drinking water

sul1, sul2, tetC, tetG, tetX, tetA, tetB, tetO, tetM, and tetW

Total concentrations of ARGs belonging to either the sulfonamide or tetracycline resistance gene class were above 105 copies mL-1

[31]

River sediments

TEM,sul1,sul2

1.09 × 10−1

~1.06 × 10−1 

[32]

Table 2. Disposal efficiency of ARGs in different biological treatments.

Processing techniques

Operating conditions

ARGs kind

Removal effect

References

Activated sludge treatment

tetO, tetW

3 logs

[202]

ermB,tetW,sul2

1.29–2.45 log (ermB), 1.13–1.62 log (tetW), 0.26–0.53 log (sul2)

[63]

CASS

tetAtetOtetWsulIsulIIblaCTX-M

>2.60±0.015 log (tetO); >2.66±0.023 log (tetW)

[203]

A/O

tet,erm,sul,qnr,bla

16.90% (total ARGs), 64.50% (tet), 92.00% (erm)

[200]

A/A/O

tet,erm,sul,qnr,bla

56.00% (tet), 70.40%–87.00% (erm)

[200]

sulI, sulII, tetO, tetW, tetQ

1.69 logs, 1.44 logs, 2.31 logs, 2.13 logs, 2.5 logs

[204]

Membrane bioreactor

sulII, tetO, tetW

2.57 logs, 7.06 logs, 6 logs

[205]

AnMBR

blandM‑1, blaCTX‑M‑15, and blaOXA‑48

2.76–3.84 logs

[206]

A/O-MBR

sulI, sulII, tetC, tetX, ereA, and int1

0.5–5.6 logs

[11]

Aerobic granular sludge

tetW,sul2,sul1,intI1,ermB

2.02 log (tetW), 1.43 log (sul2), 0.77 log (sul1), 0.55 log (intI1), 0.08 log (ermB)

[63]

Anaerobic digestion

40 °C, 56 °C, 60 °C, and 63 °C.

tetW, tetX, qnrA, and intI1.

Decreased ARGs except qnrA by 89 -96 % and ~ 99 % at 40 °C and other temperatures, and decreased qnrA by 99 % at 40, 60, and 63 °C.

[207]

MAD

35 °C, sludge retention time (SRT) 20 d.

sulI, sulII, tetA, tetO, tetX, bla , and TEM bla .

Decreased extracellular ARGs by 0.11 1.22 logs.

[208]

TAD

55 °C, SRT 20d.

sulI, sulII, tetA, tetO, tetX, bla , and TEM blaSHV.

Decreased extracellular ARGs by 0.33 1.46 logs.

[208]

Composting

Kitchen waste

tetA, tetB, tetC, tetG, tetM, tetO, tetQ, tetW, tetX, sul1, sul2, sul3, and dfrA7, qnrB, qnrS, acc(6′)-Ibcr, ermB, ermF, ermQ, ermX, mefA,

total ARGs: 99.68%–99.98% (tetracyclines: >99%; sulfonamides: 5.35%–8534.69%; quinolones: 837.30%–99.29%; macrolides: 4425.46%–98.14%)

[209]

Cattle manure

ermB, ermF, ermQ, ermX, sul1, sul2, sulA, tetA, tetB, tetC, tetE, tetG, tetK, tetM, tetO, tetQ, tetW, tetX

total ARGs: 52.69%

[210]

Sewage sludge

ermB, ermC, sul1, sul2, tetC, tetG, tetO

ermB, ermC, sul1, and tetC: 25.7%, 42.4%, 69.4%, and 44.6%, respectively

[211]

CW-surface flow

Capacity:600m/d HLR:350-450mm/d

HRT:6h

sulI, sulII, sulIII, tetA, tetB, tetC, tetE, tetH, tetM, tetO, tetW, qnrB, qnrS, and qepA

77.8% in summer, 59.5% in winter

[212]

CW-horizontal subsurface flow

Capacity: 500 m/d

intl1, sulI, sulII, dfrA, aac6, tetO, qnrA, blaNMD1, blaKPC, blaCTX, and ermB

145.6%–98.9%

[213]

CW-vertical subsurface flow

HLR: 5.1 cm/d

tet genes and intI1

33.2%–99.1%

[214]

Constructed wetland

sulI, sulII, tetO, tetW and tetQ

1.5 logs, 0.48 log, 2.1 logs, 1.5 logs, 2.1 logs

[204]

Microalgae

bla-Tem,ermB

0.56logs,1.75logs

[215]

sul1, tetQ, blaKPC, and intl1

1.2-4.9logs,2.7-6.3logs,0-1.5logs,1.2-4.8logs

[216]

(5)The section on CRISPR-Cas and nanomaterials is relatively brief. Expanding on their practical applications or incorporating recent examples would add depth and relevance.

Response: Thanks, we have added this section about CRISPR-Cas technology Line (1060-1128) as follows: ''The CRISPR-Cas antibiotic resistance gene knockout technology is a gene editing tool developed based on the bacterial adaptive immune system. The mechanism utilizes a specially engineered single-stranded guide RNA (sgRNA) to direct Cas nucleases with high precision to excise antibiotic resistance genes, thus eradicating antibiotic resistance in pathogens. CRISPR-Cas technology has made substantial progress in the domain of resistance gene removal. This system utilizes the design of specific sgRNAs to achieve precise cleavage of antibiotic resistance genes on chromosomes or plasmids. For instance, following the targeting of the bla_NDM-1 gene in Escherichia coli by Cas9, the strain exhibits a substantial increase in antibiotic sensitivity [245].……………… These challenges can be addressed by developing Cas-resistant Cas variants, optimizing AI-driven sgRNA design libraries to improve specificity, and integrating miniaturized systems (such as Cas14) with multifunctional carriers to achieve cross-species precision delivery. [253].''

Nanotechnology Line (1171-1191) as follows: “'The application of nanotechnology in the domain of environmental antibiotic resistance gene (ARG) removal has been demonstrated in a variety of scenarios. In the domain of water treatment, iron-based nano-copper bimetallic materials have been observed to form spike-like copper nanoclusters. These nanoclusters have been shown to induce oxidative stress and membrane damage in bacteria, resulting in the penetration of cells and the subsequent degradation of DNA. The removal efficiencies of intracellular and extracellular antibiotic resistance genes (ARGs) in livestock and poultry wastewater have been reported to be 3.75 and 4.36 orders of magnitude, respectively. Notably, these materials demonstrate resilience to the effects of organic pollutants, making them a promising economic solution for wastewater treatment in aquaculture. [262] In the domain of soil remediation, nitrogen-doped carbon dots (NCDs) loaded with the CRISPR-Cas9/sgRNA system have been demonstrated to exhibit precise targeting of high-risk antibiotic resistance genes (ARGs), including tet, cat, and aph(3')-Ia, within soil environments. By optimizing the ratio of NCDs to Cas9 protein, multiple genes can be edited in a simultaneous manner, thereby reducing ARG abundance by 90% within a 7-day period and overcoming the inactivation issues of traditional CRISPR components in complex environments [250]. In addition, in the context of environmental monitoring, lanthanide phosphate TbPO nanomaterials selectively adsorb eDNA with phosphate groups, achieving a 97% enrichment efficiency for extracellular ARGs in environmental samples such as tap water and river water, with a recovery rate of 78.83%, providing a highly sensitive tool for assessing the risk of antibiotic resistance gene transmission [263].''

(6)The conclusion would benefit from a clearer summary of the key findings and a stronger emphasis on critical research and policy needs.
Adding a final paragraph outlining recommendations for future research or potential interdisciplinary collaboration would further enhance the manuscript’s impact.

Response:Thanks,we have revised this section line (1242-1312) as follows: ''This review systematically reveals the closed-loop threat chain formed by antibiotics and ARGs in the environment: from the widespread release of antibiotics and ARGs from medical, agricultural, and industrial sources, to their enrichment and spread in wastewater treatment systems driven by HGT (plasmid, integron, and outer membrane vesicle-mediated cross-species transmission are key pathways) [21, ultimately posing risks to human health through exposure pathways such as the food chain and aerosols. At the molecular level, bacteria under the combined influence of various environmental stresses develop multidrug resistance primarily through co-resistance mechanisms, including altered membrane permeability, activation of efflux pumps, target site modifications, and enzymatic degradation. Additionally, biofilm resistance regulated by quorum sensing further enhances their resistance. HGT serves as the core mechanism for the dissemination of ARGs through conjugation, transformation, transduction, and outer membrane vesicles, relying on mobile genetic elements to facilitate cross-species transmission.……………… Concurrently, financial resources should be allocated to the development of an in situ monitoring network to warn of cross-border transmission of ARGs. Furthermore, efforts should be made to promote the integration and modeling of environmental-clinical resistance data to enable risk prediction from genes to population exposure.''

Reviewer 5 Report

Comments and Suggestions for Authors

All my suggestions have been well responded to and improved upon in the manustript. 

Author Response

Thank you for confirming that our revisions have adequately addressed your suggestions. We greatly appreciate your time and expertise in guiding the improvement of this manuscript.